# Interpretable dimensionality reduction of single cell transcriptome data with deep generative models

Jiarui Ding [1,2,3,4], Anne Condon [1] & Sohrab P. Shah [1,2,3,5]

Single-cell RNA-sequencing has great potential to discover cell types, identify cell states, trace development lineages, and reconstruct the spatial organization of cells. However, dimension reduction to interpret structure in single-cell sequencing data remains a challenge. Existing algorithms are either not able to uncover the clustering structures in the data or lose global information such as groups of clusters that are close to each other. We present a robust statistical model, scvis, to capture and visualize the low-dimensional structures in single-cell gene expression data. Simulation results demonstrate that low-dimensional representations learned by scvis preserve both the local and global neighbor structures in the data. In addition, scvis is robust to the number of data points and learns a probabilistic parametric mapping function to add new data points to an existing embedding. We then use scvis to analyze four single-cell RNA-sequencing datasets, exemplifying interpretable two-dimensional representations of the high-dimensional single-cell RNA-sequencing data.

[1] Department of Computer Science, University of British Columbia, Vancouver, BC V6T 1Z4, Canada. [2] Department of Molecular Oncology, BC Cancer Agency, Vancouver, BC V5Z 1L3, Canada. [3] Department of Pathology and Laboratory Medicine, University of British Columbia, Vancouver, BC V6T 2B5, Canada. [4] Present address: Broad Institute of MIT and Harvard, Cambridge, MA 02142, USA. [5] Present address: Memorial Sloan Kettering Cancer Center, 1275 York Avenue, New York, NY 10065, USA. Correspondence and requests for materials should be addressed to J.D. (email: jding@broadinstitute.org) or to S.P.S. (email: sshah@bccrc.ca)

Categorizing cell types comprising a specific organ or disease tissue is critical for comprehensive study of tissue development and function[1]. For example, in cancer, identifying constituent cell types in the tumor microenvironment together with malignant cell populations will improve understanding of cancer initialization, progression, and treatment response[2, 3]. Technical developments have made it possible to measure the DNA and/or RNA molecules in single cells by single-cell sequencing[4–15] or protein content by flow or mass cytometry[16, 17]. The data generated by these technologies enable us to quantify cell types, identify cell states, trace development lineages, and reconstruct the spatial organization of cells[18, 19]. An unsolved challenge is to develop robust computational methods to analyze large-scale single-cell data measuring the expression of dozens of protein markers to all the mRNA expression in tens of thousands to millions of cells in order to distill single-cell biology[20–23].

Single-cell datasets are typically high dimensional in large numbers of measured cells. For example, single-cell RNA-sequencing (scRNA-seq)[19, 24–26] can theoretically measure the expression of all the genes in tens of thousands of cells in a single experiment[9, 10, 14, 15]. For analysis, dimensionality reduction projecting high-dimensional data into low-dimensional space (typically two or three dimensions) to visualize the cluster structures[27–29] and development trajectories[30–33] is commonly used. Linear projection methods such as principal component analysis (PCA) typically cannot represent the complex structures of single-cell data in low dimensional spaces. Nonlinear dimension reduction, such as the $t$-distributed stochastic neighbor embedding algorithm (t-SNE)[34–39], has shown reasonable results for many applications and has been widely used in single-cell data processing[1, 40, 41]. However, t-SNE has several limitations[42]. First, unlike PCA, it is a non-parametric method that does not learn a parametric mapping. Therefore, it is not natural to add new data to an existing t-SNE embedding. Instead, we typically need to combine all the data together and rerun t-SNE. Second, as a non-parametric method, the algorithm is sensitive to hyperparameter settings. Third, t-SNE is not scalable to large datasets because it has a time complexity of $O(N^2D)$ and space complexity of $O(N^2)$, where $N$ is the number of cells and $D$ is the number of expressed genes in the case of scRNA-seq data. Fourth, t-SNE only outputs the low-dimensional coordinates but without any uncertainties of the embedding. Finally, t-SNE typically preserves the local clustering structures very well given proper hyperparameters, but more global structures such as a group of subclusters that form a big cluster are missed in the low-dimensional embedding.

In this paper, we introduce a robust latent variable model, scvis, to capture underlying low-dimensional structures in scRNA-seq data. As a probabilistic generative model, our method learns a parametric mapping from the high-dimensional space to a low-dimensional embedding. Therefore, new data points can be directly added to an existing embedding by the mapping function. Moreover, scvis estimates the uncertainty of mapping a high-dimensional point to a low-dimensional space that adds rich capacity to interpret results. We show that scvis has superior distance preserving properties in its low-dimensional projections leading to robust identification of cell types in the presence of noise or ambiguous measurements. We extensively tested our method on simulated data and several scRNA-seq datasets in both normal and malignant tissues to demonstrate the robustness of our method.

## Results

### Modeling and visualizing scRNA-seq data.
Although scRNA-seq datasets have high dimensionality, their intrinsic dimensionalities are typically much lower. For example, factors such as cell type and patient origin explain much of the variation in a study of metastatic melanoma[3]. We therefore assume that for a high-dimensional scRNA-seq dataset $\mathcal{D} = \{\mathbf{x}_n\}_{n=1}^N$ with $N$ cells, where $\mathbf{x}_n$ is the expression vector of cell $n$, the $\mathbf{x}_n$ distribution is governed by a latent low-dimensional random vector $\mathbf{z}_n$ (Fig. 1a). For visualization purposes, the dimensionality $d$ of $\mathbf{z}_n$ is typically two or three. We assume that $\mathbf{z}_n$ is distributed according to a prior, with the joint distribution of the whole model as $p(\mathbf{z}_n \mid \boldsymbol{\theta})p(\mathbf{x}_n \mid \mathbf{z}_n, \boldsymbol{\theta})$. For simplicity, we can choose a factorized standard normal distribution for the prior $p(\mathbf{z}_n \mid \boldsymbol{\theta}) = \prod_{i=1}^d \mathcal{N}(z_{n,i} \mid 0, \mathbf{I})$. The distribution $p(\mathbf{x}_n \mid \boldsymbol{\theta}) = \int p(\mathbf{z}_n \mid \boldsymbol{\theta})p(\mathbf{x}_n \mid \mathbf{z}_n, \boldsymbol{\theta})\mathrm{d}\mathbf{z}_n$ can be a complex multimodal high-dimensional distribution. To represent complex high-dimensional distributions, we assume that $p(\mathbf{x}_n \mid \mathbf{z}_n, \boldsymbol{\theta})$ is a location-scale family distribution with location parameter $\boldsymbol{\mu}_{\boldsymbol{\theta}}(\mathbf{z}_n)$ and scale parameter $\boldsymbol{\sigma}_{\boldsymbol{\theta}}(\mathbf{z}_n)$; both are functions of $\mathbf{z}_n$ parameterized by a neural network with parameter $\boldsymbol{\theta}$. The inference problem is to compute the posterior distribution $p(\mathbf{z}_n \mid \mathbf{x}_n, \boldsymbol{\theta})$, which is however intractable to compute. We therefore use a variational distribution $q(\mathbf{z}_n \mid \mathbf{x}_n, \boldsymbol{\phi})$ to approximate the posterior (Fig. 1b). Here $q(\mathbf{z}_n \mid \mathbf{x}_n, \boldsymbol{\phi})$ is a multivariate normal distribution with mean $\boldsymbol{\mu}_{\boldsymbol{\phi}}(\mathbf{x}_n)$ and standard deviation $\boldsymbol{\sigma}_{\boldsymbol{\phi}}(\mathbf{x}_n)$. Both parameters are (continuous) functions of $\mathbf{x}_n$ parameterized by a neural network with parameter $\boldsymbol{\phi}$. To model the data distribution well (with a high likelihood of $\int p(\mathbf{z}_n \mid \boldsymbol{\theta})p(\mathbf{x}_n \mid \mathbf{z}_n, \boldsymbol{\theta})\mathrm{d}\mathbf{z}_n$), the model tends to assign similar posterior distributions $p(\mathbf{z}_n \mid \mathbf{x}_n, \boldsymbol{\theta})$ to cells with similar expression profiles. To explicitly encourage cells with similar expression profiles to be proximal (and those with dissimilar profiles to be distal) in the latent space, we add the t-SNE objective function on the latent $\mathbf{z}$ distribution as a constraint. More details about the model and the inference algorithms are presented in the Methods section. The scvis model is implemented in Python using Tensorflow[43] with a command-line interface and is freely available from https://bitbucket.org/jerry00/scvis-dev.

### Single-cell datasets.
We analyzed four scRNA-seq datasets in this study[1, 3, 9, 44]. Data were mostly downloaded from the single-cell portal[45]. Two of these datasets were originally used to study intratumor heterogeneity and the tumor microenvironment in metastatic melanoma[3] and oligodendroglioma[44], respectively. One dataset was used to categorize the mouse bipolar cell populations of the retina[1], and one dataset was used to categorize all cell types in the mouse retina[9]. For all the scRNA-seq datasets, we used PCA (as a noise-reduction preprocessing step[1, 19]) to project the cells into a 100-dimensional space and used the projected coordinates in the 100-dimensional spaces as inputs to scvis. We also used two mass cytometry (CyTOF) datasets consisting of bone marrow mononuclear cells from two healthy adult donors H1 and H2[17]. For CyTOF data, since their dimensionality (32) is relatively low, we directly used these data as inputs to scvis.

### Experimental setting and implementation.
The variational approximation neural network has three hidden layers ($l_1$, $l_2$, and $l_3$) with 128, 64, and 32 hidden units each, and the model neural network has five hidden layers ($l_1'$, $l_2'$, $l_3'$, $l_4'$, and $l_5'$) with 32, 32, 32, 64, and 128 units each. We use the exponential linear unit activation function as it has been shown to speed up the convergence of optimization[46] and the Adam stochastic optimization algorithm with a learning rate of 0.01[47]. Details about the influence of these hyperparameters on results are presented in the Methods section. The time complexity to compute the t-SNE loss is quadratic in terms of the number of data points. Consequently, we use mini-batch optimization and set the mini-batch size to 512 (cells). We expect that a large batch of data could be better in

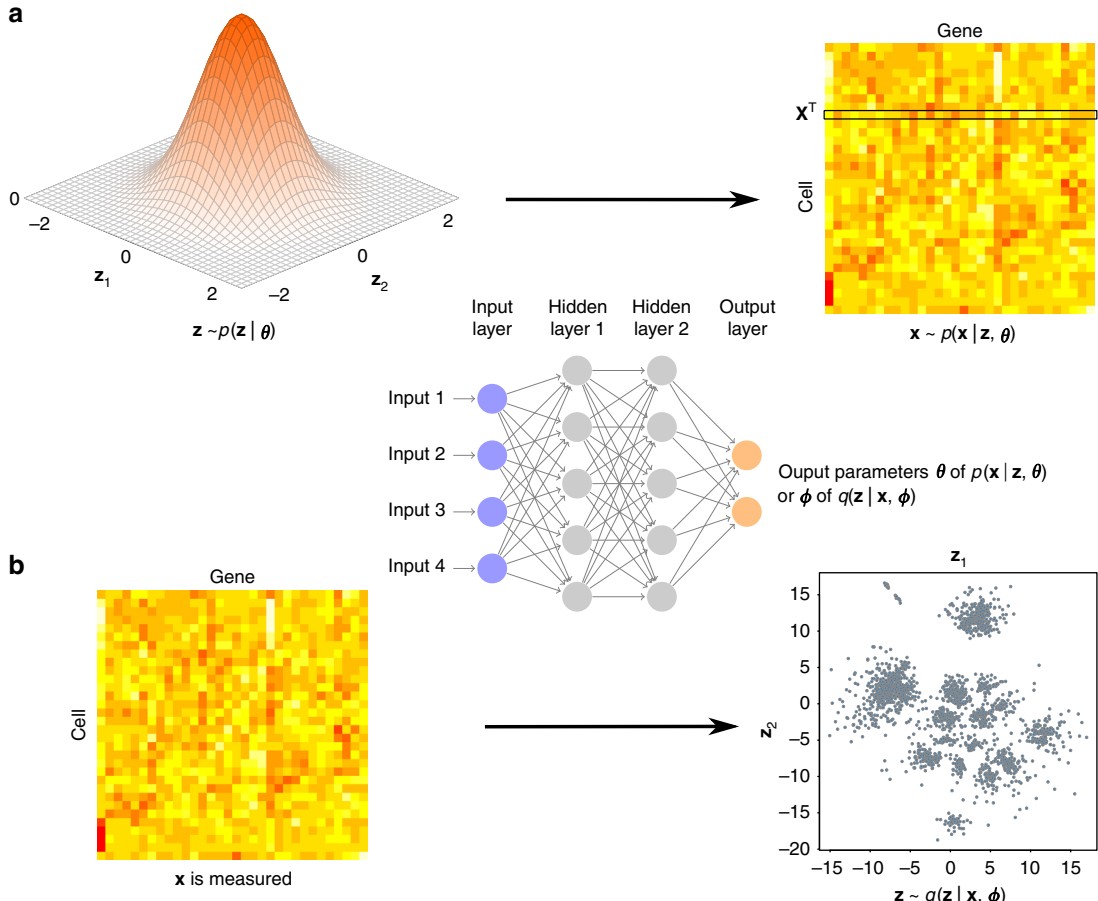

**Fig. 1** Overview of the scvis method. **a** scvis model assumptions: given a low-dimensional point drawn from a simple distribution, e.g., a two-dimensional standard normal distribution, a high-dimensional gene expression vector of a cell can be generated by drawing a sample from the distribution $p(\mathbf{x} \mid \mathbf{z}, \boldsymbol{\theta})$. The heatmap represents a cell–gene expression matrix, where each row is a cell and each column is a gene. Color encodes the expression levels of genes in cells. The data-point-specific parameters $\boldsymbol{\theta}$ are determined by a model neural network. The model neural network (a feedforward neural network) consists of an input layer, several hidden layers, and an output layer. The output layer outputs the parameters $\boldsymbol{\theta}$ of $p(\mathbf{x} \mid \mathbf{z}, \boldsymbol{\theta})$. **b** scvis inference: given a high-dimensional gene expression vector of a cell (a row of the heatmap), scvis obtains its low-dimensional representation by sampling from the conditional distribution $q(\mathbf{z} \mid \mathbf{x}, \boldsymbol{\phi})$. The data-point-specific parameters $\boldsymbol{\phi}$ are determined by a variational inference neural network. The inference neural network is also a feedforward neural network and its output layer outputs the parameters $\boldsymbol{\phi}$ of $q(\mathbf{z} \mid \mathbf{x}, \boldsymbol{\phi})$. Again, the heatmap represents a cell–gene expression matrix. The scatter plot shows samples drawn from the variational posterior distributions $q(\mathbf{z} \mid \mathbf{x}, \boldsymbol{\phi})$

estimating the high-dimensional data manifold, however we found that 512 cells work accurately and efficiently in practice. We run the Adam stochastic gradient descent algorithm for 500 epochs for each dataset with at least 3000 iterations by default. For large datasets, running 500 epochs is computationally expensive, we therefore run the Adam algorithm for a maximum of 30,000 iteration or two epochs (which ever is larger). We use an L2 regularizer of 0.001 on the weights of the neural networks to prevent overfitting.

**Benchmarking scvis against t-SNE on simulated data**. To demonstrate that scvis can robustly learn a low-dimensional representation of the input data, we first simulated data in a two-dimensional space (for easy visualization) as in Fig. 2a. The big cluster on the left consisted of 1000 points and the five small clusters on the right each had 200 points. The five small clusters were very close to each other and could roughly be considered as a single big cluster. There were 200 uniformly distributed outliers around these six clusters. For each two-dimensional data point with coordinates $(x, y)$, we then mapped it into a nine-dimensional space by the transformation $(x+y, x-y, xy, x^2,$

$y^2, x^2y, xy^2, x^3, y^3)$. Each of the nine features was then divided by its corresponding maximum absolute value.

Although t-SNE (with default parameter setting, we used the efficient Barnes-Hut t-SNE[34] R wrapper package[48]) uncovered the six clusters in this dataset, it was still challenging to infer the overall layout of the six clusters (Fig. 2b). t-SNE by design preserves local structure of the high-dimensional data, but the "global" structure is not reliable. Moreover, for the uniformly distributed outliers, t-SNE put them into several compact clusters, which were adjacent to other genuine clusters.

The scvis results, on the other hand, better preserved the overall structure of the original data (Fig. 2c): (1) The five small clusters were on one side, and the big cluster was on the other side. The relative positions of the clusters were also preserved. (2) Outliers were scattered around the genuine clusters as in the original data. In addition, as a probabilistic generative model, scvis not only learned a low-dimensional representation of the input data but also provided a way to quantify the uncertainty of the low-dimensional mapping of each input data point by its log-likelihood. We colored the low-dimensional embedding of each data point by its log-likelihood (Fig. 2d). We can see that generally scvis put most of its modeling power to model the five

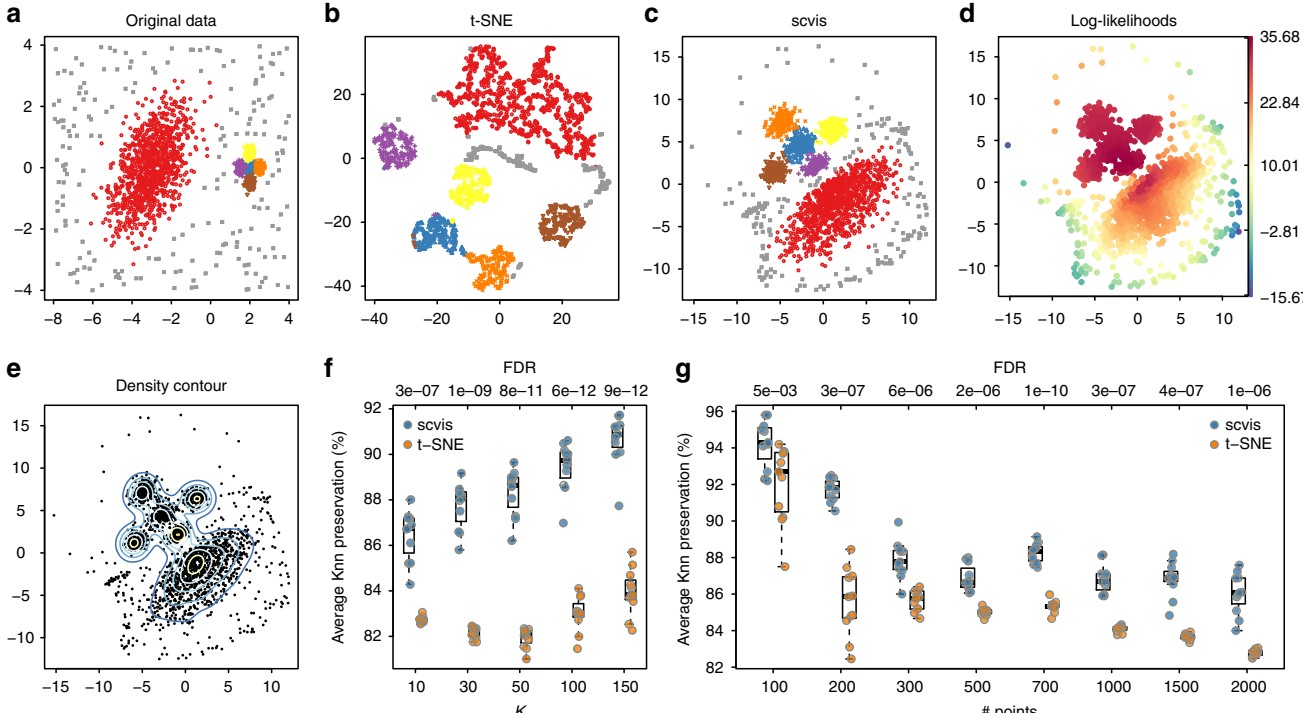

**Fig. 2** Benchmarking scvis against t-SNE on synthetic data. **a** The original 2200 two-dimensional synthetic data points (points are colored by cluster labels. The randomly distributed outliers are also colored in a distinct color, same for **b**, **c**), **b** t-SNE results on the transformed nine-dimensional dataset with default perplexity parameter of 30, **c** scvis results, **d** coloring points based on their log-likelihoods from scvis, **e** the kernel density estimates of the scvis results, where the density contours are represented by lines, **f** average $K$-nearest neighbor preservations across ten runs for different $K$s, and **g** the average $K$-nearest neighbor preservations ($K = 10$) for different numbers of subsampled data points from ten repeated runs. The numbers at the top are the adjusted $p$-values (FDR, one-sided Welch's $t$-test) comparing the average $K$nn preservations from scvis with those from t-SNE. Boxplots in **f**, **g** denote the medians and the interquartile ranges (IQRs). The whiskers of a boxplot are the lowest datum still within 1.5 IQR of the lower quartile and the highest datum still within 1.5 IQR of the upper quartile

compact clusters, while the outliers far from the five compact clusters tended to have lower log-likelihoods. Thus, by combining the log-likelihoods and the low-dimensional density information (Fig. 2e), we can better interpret the structure in the original data and uncertainty over the projection.

The low-dimensional representation may change for different runs because the scvis objective function can have different local maxima. To test the stability of the low-dimensional representations, we ran scvis ten times. Generally, the two-dimensional representations from the ten runs (Supplementary Fig. 1a–j) showed similar patterns as in Fig. 2c. As a comparison, we also ran t-SNE ten times, and the results (Supplementary Fig. 1k–t) showed that the layouts of the clusters were less preserved, e.g., the relative positions of the clusters changed from run to run. To quantitatively compare scvis and t-SNE results, we computed the average $K$-nearest neighbor ($K$nn) preservations across runs for $K \in \{10, 30, 50, 100, 150\}$. Specifically, for the low-dimensional representation from each run, we constructed $K$nn graphs for different $K$s. We then computed the $K$nn graph from the high-dimensional data for a specific $K$. Finally, we compared the average overlap of the $K$nn graphs from the low-dimensional representations with the $K$nn graph from the high-dimensional data for a specific $K$. For scvis, the median $K$nn preservations monotonically increased from 86.7% for $K = 10$, to 90.9% for $K = 150$ (Fig. 2f). For t-SNE, the median $K$nn preservations first decreased from 82.7% for $K = 10$ to 82.1% for $K = 50$ (consistent with t-SNE preserving local structures) and then increased to 84.0% for $K = 150$. Thus scvis preserved $K$nn more effectively than t-SNE.

To test how scvis performs on smaller datasets, we subsampled the nine-dimensional synthetic dataset. Specifically, we subsampled 100, 200, 300, 500, 700, 1000, 1500, and 2000 points from the original dataset and ran scvis 11 times on each subsampled dataset. We then computed the $K$nn preservations ($K = 10$) and found that the $K$nn preservations from the scvis results were significantly higher than those from t-SNE results (false discovery rate (FDR) <0.01 for all the subsampled datasets, one-sided Welch's $t$-test, Fig. 2g). scvis performs very well on all the subsampled datasets (Supplementary Fig. 2a–h). Even with just 100 data points, the two-dimensional representation (Supplementary Fig. 2a) preserved much of the structure in the data. The log-likelihoods estimated from the subsampled data also recapitulated the log-likelihoods from the original 2200 data points (Supplementary Fig. 3a–h). The t-SNE results on the subsampled datasets (Supplementary Fig. 2i–p) generally revealed the clustering structures. However, the relative positions of the five clusters and the big cluster were largely inaccurate.

To test the performance of scvis when adding new data to an existing embedding, we increased by tenfold the number of points in each cluster and the number of outliers (for a total of 22,000 points) using a different random seed. The embedding (Fig. 3a, b) was very similar to that of the 2200 training data points in Fig. 2c, d. We trained $K$nn classifiers on the embedding of the 2200 training data for $K \in \{5, 9, 17, 33, 65\}$ and used the trained classifiers to classify the embedding of the 22,000 points, repeating 11 times. Median accuracy (the proportion of points correctly assigned to their corresponding clusters) was 98.1% for $K = 5$ and 94.8% for $K = 65$. The performance decreased mainly

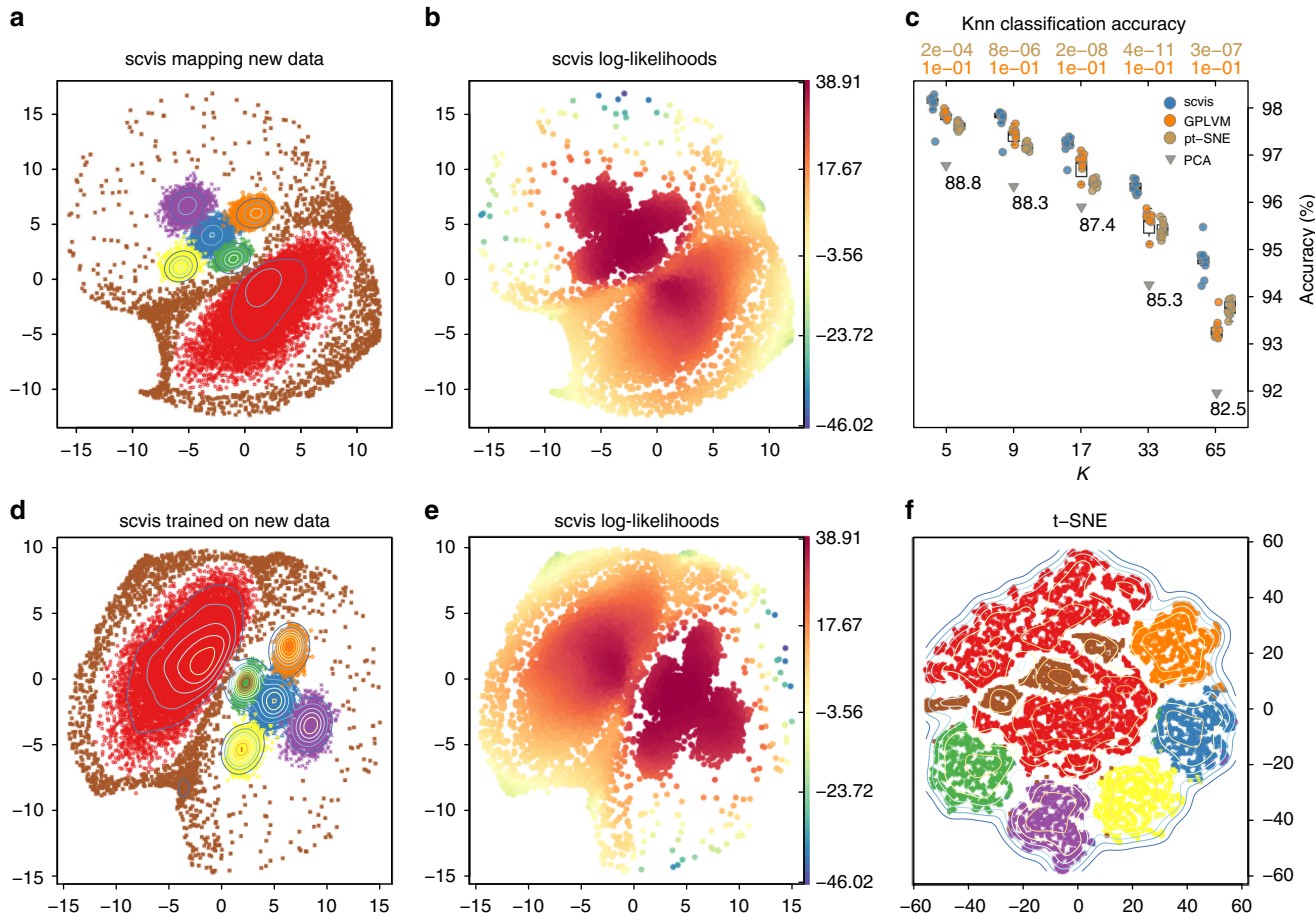

**Fig. 3** Benchmarking scvis against GPLVM, parametric t-SNE, and PCA to embed 22,000 synthetic out-of-sample data. **a** scvis mapping 22,000 new data points based on the learned probabilistic mapping function from the 2200 training data points, **b** the estimated log-likelihoods, and **c** the average $K$-nearest neighbor classification accuracies for different $K$s across 11 runs, the classifiers were trained on the 11 embeddings from the 2200 points. The numbers at the top are the FDR (one-sided Mann–Whitney $U$-test) comparing the $K$-nearest neighbor classification accuracy from scvis with those from GPLVM (orange, bottom) and those from parametric t-SNE (golden, top). Notice that, for GPLVM, two runs produced bad results and were not plotted in the figure. Boxplots denote the medians and the interquartile ranges (IQR). The whiskers of a boxplot are the lowest datum still within 1.5 IQR of the lower quartile and the highest datum still within 1.5 IQR of the upper quartile. **d** scvis results on the larger dataset with the same perplexity parameter as used in Fig. 2; **e** scvis log-likelihoods on the larger dataset; and **f** t-SNE results on the larger dataset

because, for a larger $K$, the outliers were wrongly assigned to the six genuine clusters.

We then benchmarked scvis against Gaussian process latent variate model[49] (GPLVM, implemented in the GPy[50] package), parametric t-SNE[51] (pt-SNE), and PCA on embedding the 22,000 out-of-sample data points. We used the 11 scvis models trained on the small nine-dimensional synthetic dataset with 2200 data points to embed the larger nine-dimensional synthetic data with 22,000 data points. Similarly, we trained 11 GPLVM models and pt-SNE models on the small nine-dimensional synthetic dataset and applied these models to the bigger synthetic dataset. To compare the abilities of the trained models to embed unseen data, we trained $K$nn classifiers on the two-dimensional representations (of the small 2200 data points) outputted from different algorithms. These $K$nn classifiers were used to classify the two-dimensional coordinates of the 22,000 data points outputted from different algorithms. scvis was significantly better than GPLVM and pt-SNE for different $K$s (Fig. 3c, two runs of GPLVM produced bad results and were not plotted in the figure, FDR < 0.05, one-sided Mann–Whitney $U$-test). For PCA, because the model is unique for a given dataset, we generated unique two-dimensional coordinates for the 22,000 out-of-sample data points. The $K$nn classifiers trained on the PCA coordinates were worse

than those from scvis, GPLVM, and pt-SNE in terms of the mean classification accuracies for different Ks.

As a non-parametric dimension reduction method, t-SNE was sensitive to hyperparameter setting, especially the perplexity parameter (the effective number of neighbors, see the Methods section for details). The optimal perplexity parameter increased as the total number of data points increased. In contrast, as we adopted mini-batch for training scvis by subsampling, e.g., 512 cells each time, scvis was less sensitive to the perplexity parameter as we increase the total number of training data points because the number of cell is fixed at 512 at each training step. Therefore, scvis performed well on approximately an order of magnitude larger dataset (Fig. 3d, e), without changing the perplexity parameter for scvis. For this larger dataset, the t-SNE results (Fig. 3f) were difficult to interpret without the ground-truth cluster information, because it was already difficult to see how many clusters in this dataset, not to mention to uncover the overall structure of the data. Although by increasing the perplexity parameter, the performance of t-SNE became better (Supplementary Fig. 4), the outliers still formed distinct clusters, and it remains difficult to set this parameter in practice.

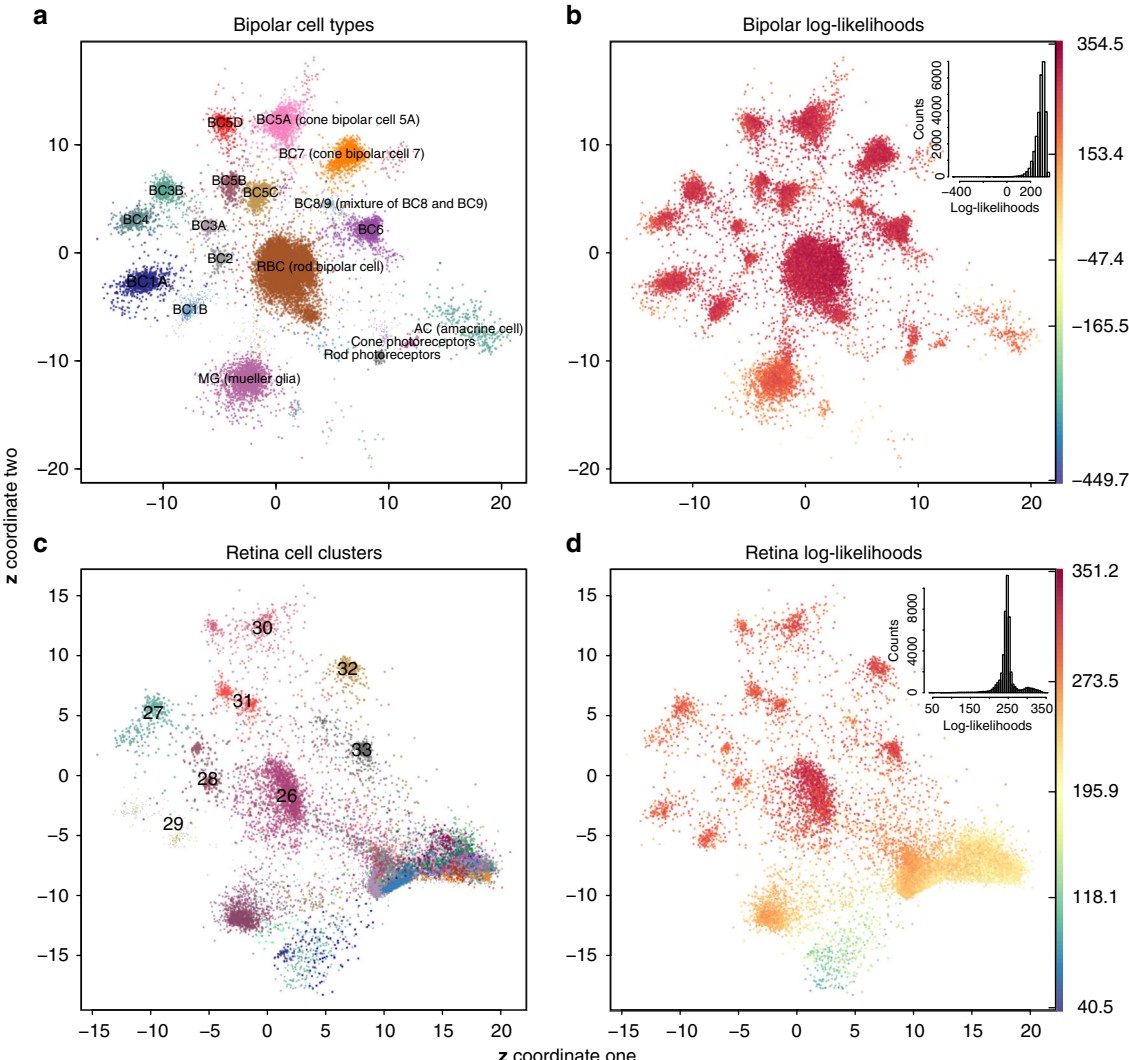

**Fig. 4** Learning a probabilistic mapping function from the bipolar data and applying the function to the independently generated mouse retina dataset. **a** scvis learned two-dimensional representations of the bipolar dataset, **b** coloring each point by the estimated log-likelihood, **c** the whole mouse retina dataset was directly projected to a two-dimensional space by the probabilistic mapping function learned from the bipolar data, and **d** coloring each point from the retina dataset by the estimated log-likelihood

**Learning a parametric mapping for a single-cell dataset**. We next analyzed the scvis learned probabilistic mapping from a training single-cell dataset and tested how it performed on unseen data. We first trained a model on the mouse bipolar cell of the retina dataset[1] and then used the learned model to map the independently generated mouse retina dataset[9]. The two-dimensional coordinates from the bipolar dataset captured much information in this dataset (Fig. 4a). For example, non-bipolar cells such as amacrine cells, Mueller glia, and photoreceptors were at the bottom, the rod bipolar cells were in the middle, and the cone bipolar cells were on the top left around the rod bipolar cells. Moreover, the "OFF" cone bipolar cells (BC1A, BC1B, BC2, BC3A, BC3B, BC4) were on the left and close to each other, and the "ON" cone bipolar cells (BC5A-D, BC6, BC7, BC8/9) were at the top. Cell doublets and contaminants (accounting for 2.43% of the cells comprised eight clusters[1], with distinct color and symbol combinations in Fig. 4a but not labeled) were rare in the bipolar datasets, and they were mapped to low-density regions in the low-dimensional plots (Fig. 4a).

Consistent with the synthetic data (Fig. 2), t-SNE put the "outlier" cell doublets and contaminants into very distinct compact clusters (Supplementary Fig. 5a, t-SNE coordinates

from Shekhar et al.[1]). In addition, although t-SNE mapped cells from different cell populations into distinct regions, more global organizations of clusters of cells were missed in the t-SNE embedding. The "ON" cone bipolar cell clusters, the "OFF" cone bipolar cell clusters, and other non-bipolar cell clusters were mixed together in the t-SNE results.

The bipolar cells tended to have higher log-likelihoods than non-bipolar cells such as amacrine cells, Mueller glia, and photoreceptors (Fig. 4b), suggesting that the model used most of its power to model the bipolar cells, while other cell types were not modeled as well. The embedded figure at the top right corner shows the histogram of the log-likelihoods. The majority of the points exhibited high log-likelihoods (with a median of 292.4). The bipolar cells had significantly higher log-likelihoods (median log-likelihood of 298.4) relative to non-bipolar cells (including amacrine cells, Mueller glia, rod and cone photoreceptors) (median log-likelihood of 223.6; one-sided Mann–Whitney U-test FDR < 0.001; Supplementary Fig. 5b). The amacrine cells had the lowest median log-likelihood (median log-likelihood for amacrine cells, Mueller glia, rod and cone photoreceptors were 226.4, 187.3, 222.7, and 205.4, respectively; Supplementary Fig. 5b).

We benchmarked scvis against GPLVM, pt-SNE, and PCA on embedding out-of-sample scRNA-seq data, performing a five-fold cross-validation analysis on the bipolar dataset. Specifically, we partitioned the bipolar dataset into five roughly equal size subsamples and held out one subsample as out-of-sample evaluation data, using the remaining four subsamples as training data to learn different models. We then trained $K$nn classifiers on the two-dimensional representations of the training data and then used the $K$nn classifiers to classify the two-dimensional representations of the out-of-sample evaluation data. The process was repeated five times with each of the five subsamples used exactly once as the out-of-sample validation data. scvis was significantly better than pt-SNE, GPLVM, and PCA on embedding the out-of-samples (Supplementary Fig. 6a, b, FDR < 0.05, one-sided Welch's $t$-test).

We used the learned probabilistic mapping from the bipolar cells to map the independent whole-retina dataset[9]. We first projected the retina dataset to the subspace spanned by the first 100 principal direction vectors of the bipolar dataset and then mapped each 100-dimensional vector to a two-dimensional space based on the learned scvis model from the bipolar dataset. The bipolar cell clusters in the retina dataset identified in the original study[9] (clusters 26–33) tended to be mapped to the corresponding bipolar cell subtype regions discovered in the study[1] (Fig. 4c). Although Macosko et al.[9] only identified eight subtypes of bipolar cells, all the recently identified 14 subtypes of bipolar cells[1] were possibly present in the retina dataset as can be seen from Fig. 4c, i.e., cluster 27 (BC3B and BC4), cluster 28 (BC2 and BC3A), cluster 29 (BC1A and BC1B), cluster 30 (BC5A and BC5D), cluster 31 (BC5B and BC5C), and cluster 33 (BC6 and BC8/9).

Interestingly, there was a cluster just above the rod photoreceptors (Fig. 4c) consisting of different subtypes of bipolar cells. In the bipolar dataset, cell doublets or contaminants were mapped to this region (Fig. 4a). We used densitycut[52] to cluster the two-dimensional mapping of all the bipolar cells from the retina dataset to detect this mixture of bipolar cell cluster (Supplementary Fig. 5c, where the 1535 high-density points in this cluster were labeled with red circles). To test whether this mixture cell population was an artifact of the projection, we randomly drew the same number of data points from each bipolar subtype as in the mixture cluster and computed the $K$nns of each data point (here $K$ was set to $\log_2(1535) = 11$). We found that the 11 nearest neighbors of the points from the mixture clusters were also mostly from the mixture cluster (median of 11 and mean of 10.8), while for the randomly selected points from the bipolar cells, a relatively small number of points of their 11 nearest neighbors (median of 0 and mean of 0.2) were from the mixture cluster. The results suggest that the bipolar cells in the mixture cluster were substantially different from other bipolar cells. Finally, this mixture of bipolar cells had significantly lower log-likelihoods compared with other bipolar cells (one-sided Mann–Whitney $U$-test $p$-value <0.001, Supplementary Fig. 5d).

Non-bipolar cells, especially Mueller glia cells, were mapped to the corresponding regions as in the bipolar dataset (Fig. 4c). Photoreceptors (rod and cone photoreceptors accounting for 65.6 and 4.2% of all the cells from the retina[9]) were also mapped to their corresponding regions as in the bipolar dataset (Supplementary Fig. 5e). The amacrine cells (consisting of 21 clusters) together with horizontal cells and retinal ganglion cells were mapped to the bottom right region (Supplementary Fig. 5f); all the amacrine cells were assigned the same label and the same color.

As in the training bipolar data, the bipolar cells in the retina dataset also tended to have high log-likelihoods, and other cells tended to have relatively lower log-likelihoods (Fig. 4d). The embedded plot on the top right corner shows a bimodal distribution of the log-likelihoods. The "Other" cells types (horizontal cells, retina ganglion cells, microglia cells, etc) that were only in the retina dataset had the lowest log-likelihoods (median log-likelihoods of 181.7, Supplementary Fig. 5d).

It is straightforward to project scRNA-seq to a higher than two-dimensional space. To evaluate how scvis performs on higher-dimensional maps, we projected the bipolar data to a three-dimensional space. We obtained better average log-likelihood per data point, i.e., 255.1 versus 253.3 (from the last 100 iterations) by projecting the data to a three-dimensional space compared to projecting the data to a two-dimensional space (Supplementary Fig. 7). In addition, the average $\mathbb{KL}$ divergence was smaller (2.7 versus 4.1 from the last 100 iterations) by projecting the data to a three-dimensional space.

Finally, to demonstrate that scvis can be used for other types of single-cell data, we learned a parametric mapping from the CyTOF data H2 and then directly used the mapping to project the CyTOF data H1 to a two-dimensional space. As can be seen from Supplementary Fig. 8a, all the 14 cell types were separated (although CD16+ and CD16− NK cells have some overlaps), and CD4 T cells and CD8 T cells clusters are adjacent to each other. Moreover, the high quality of the mapping carried over to the CyTOF data H1 (72,463 cells, Supplementary Fig. 8a, b).

**Tumor microenvironments and intratumor heterogeneity.** We next used scvis to analyze tumor microenvironments and intratumor heterogeneity. The oligodendroglioma dataset consists of mostly malignant cells (Supplementary Fig. 9a). We used densitycut[52] to cluster the two-dimensional coordinates to produce 15 clusters (Supplementary Fig. 9b). The non-malignant cells (microglia/macrophage and oligodendrocytes) formed two small clusters on the left and each consisted of cells from different patients. We therefore computed the entropy of each cluster based on the cells of origin (enclosed bar plot). As expected, the non-malignant clusters (cluster one and cluster five) had high entropies. Cluster 12 (cells mostly from MGH53 and MGH54) and cluster 14 (cells from MGH93 and MGH94) also had high entropies (Fig. 5a). The cells in these two clusters consisted of mostly astrocytes (Fig. 5b; the oligodendroglioma cells could roughly be classified as oligodendrocyte, astrocyte, or stem-like cells.) Interestingly, cluster 15 had the highest entropy, and these cells had significant higher stem-like scores (one-sided Welch's $t$-test $p$-value <$10^{-12}$). We also colored cells by the cell-cycle scores (G1/S scores, Supplementary Fig. 9c; G2/M scores, Supplementary Fig. 9d) and found that these cells also had significantly higher G1/S scores (one-sided Welch's $t$-test $p$-value <$10^{-12}$) and G2/M scores (one-sided Welch's $t$-test $p$-value <$10^{-9}$). Therefore, cluster 15 cells consisted of mostly stem-like cells, and these cells were cycling.

Malignant cells formed distinct clusters even if they were from the same patient (Fig. 5a). We next colored each malignant cell by its lineage score[44] (Fig. 5b). The cells in some clusters highly expressed the astrocyte gene markers or the oligodendrocyte gene markers. The stem-like cells tended to be rare and they could link "outliers" connecting oligodendrocyte and astrocyte cells in the two-dimensional scatter plots (Fig. 5b). In addition, some clusters of cells consisted of mixtures of cells (e.g., both oligodendrocyte and stem-like cells), suggesting that other factors such as genetic mutations and epigenetic measurements would be required to fully interpret the clustering structures in the dataset.

For the melanoma dataset, the authors profiled both malignant cells and non-malignant cells[3]. The malignant cells originated from different patients were mapped to the bottom left region (Fig. 5c). These malignant cells were further subdivided by the patients of origin (Fig. 5d). Similar to the oligodendroglioma

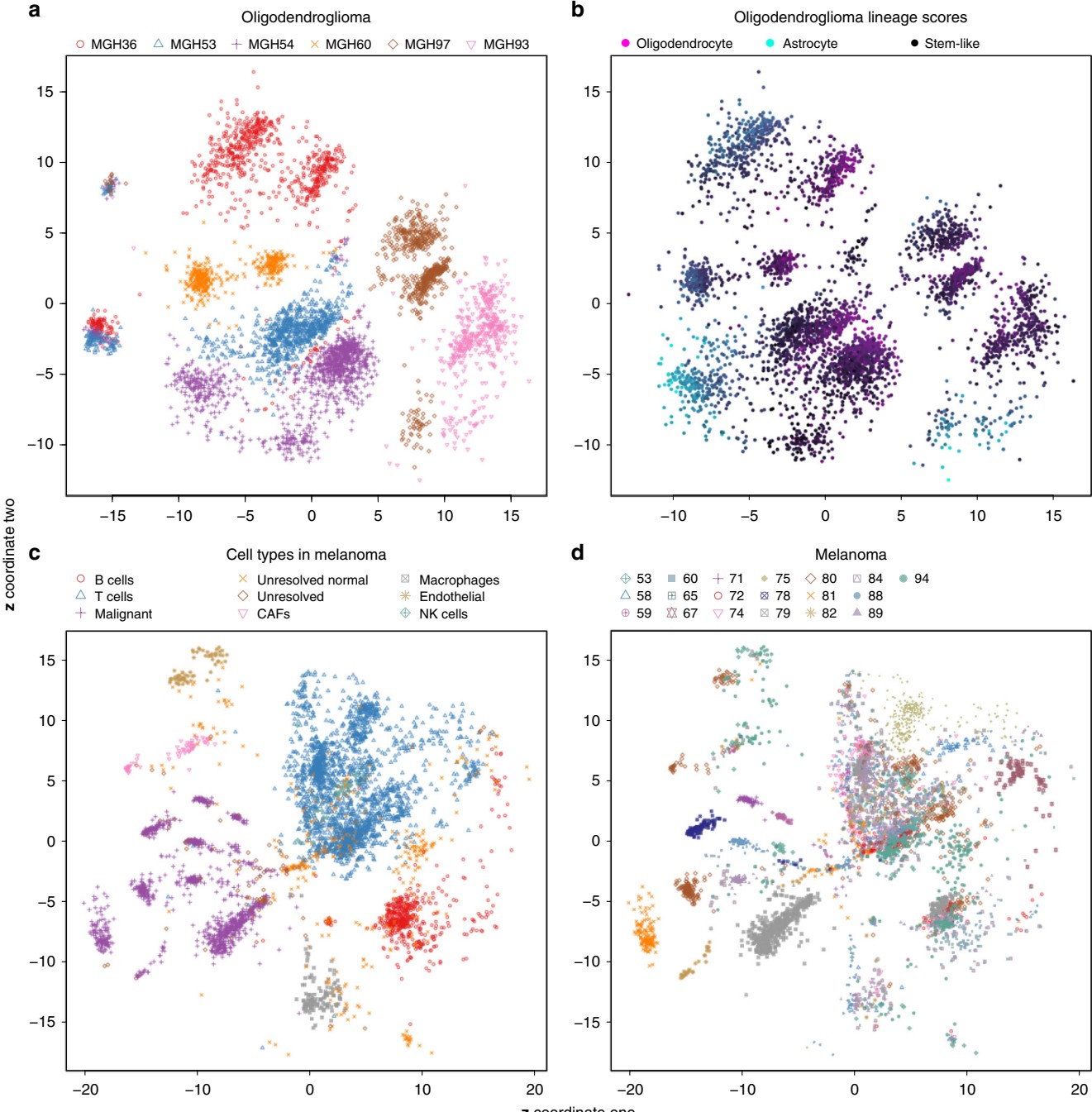

**Fig. 5** scvis learned low-dimensional representations. **a** The oligodendroglioma dataset, each cell is colored by its patient of origin, **b** the oligodendroglioma dataset, each cell is colored by its linage score from Tirosh et al.[44], **c** the melanoma dataset, each cell is colored by its cell type, and **d** the melanoma dataset, each cell is colored by its patient of origin

dataset, non-malignant immune cells such as T cells, B cells, and macrophages, even from different patients, tended to be grouped together by cell types instead of patients of origin of the cells (Fig. 5c, d), although for some patients (e.g., 75, 58, and 67, Fig. 5d), their immune cells showed patient-specific bias. We did a differential expression analysis of patient 75 T cells and other patient T cells using limma[53]. Most of the top 100 differently expressed genes were ribosome genes (Supplementary Fig. 10a), suggesting that batch effects could be detectable between patient 75 T cells and other patient T cells.

Interestingly, as non-malignant cells, cancer-associated fibroblasts (CAFs) were mapped to the region adjacent to the malignant cells. The endothelial cells were just above the CAFs (Fig. 5d). To test whether these cells were truly more similar with the malignant cells than with immune cells, we first computed the average principal component values in each type of cells and did a hierarchical clustering analysis (Supplementary Fig. 10b). Generally, there were two clusters: one cluster consisted of the immune cells and the "Unsolved normal" cells, while the other cluster consisted of CAFs, endothelial cells, malignant cells, and the "Unsolved" cells, indicating CAFs and endothelial cells were

more similar to malignant cells (they had high PC1 values) than to the immune cells.

## Discussion

We have developed a novel method, scvis, for modeling and reducing dimensionality of single-cell gene expression data. We demonstrated that scvis can robustly preserve the structures in high-dimensional datasets, including in datasets with small numbers of data points.

Our contribution has several important implications for the field. As a probabilistic generative model, scvis provides not only the low-dimensional coordinate for a given data point but also the log-likelihood as a measure of the quality of the embedding. The log-likelihoods could potentially be used for outlier detection, e.g., for the bipolar cells in Fig. 4b, the log-likelihood histogram shows a long tail of data points with relatively low log-likelihoods, suggesting some outliers in this dataset (the non-bipolar cells). The log-likelihoods could also be useful in mapping new data. For example, although horizontal cells and retinal ganglion cells were mapped to the region adjacent to/overlap the region occupied by amacrine cells, these cells exhibited low log-likelihoods, suggesting that further analyses were required to elucidate these cell types/subtypes.

scvis preserves the "global" structure in a dataset, greatly enhancing interpretation of projected structures in scRNA-seq data. For example, in the bipolar dataset, the "ON" bipolar cells were close to each other in the two-dimensional representation in Fig. 4a, and similarly, the "OFF" bipolar cells were close to each other. For the oligodendroglioma dataset, the cells can be first divided into normal cells and malignant cells. The normal cells formed two clusters, with each cluster of cells consisting of cells from multiple patients. The malignant cells, although from the same patient, formed multiple clusters with cell clusters from the same patient adjacent to each other. Adjacent malignant cell clusters from different patients tended to selectively express the oligodendrocyte marker genes or the astrocyte marker genes. For the metastatic melanoma dataset, malignant cells from different patients, although mapped to the same region, formed clusters based on the patient origin of the cells, while immune cells from different patients tended to be clustered together by cell types. From the low-dimensional representations, we can hypothesize that the CAFs were more "similar" to the malignant cells than to the immune cells.

Other methods, e.g., the SIMLR algorithm, improve the t-SNE algorithm[54] by learning a similarity matrix between cells, and the similarity matrix is used as the input of t-SNE for dimension reduction. However, SIMLR is computationally expensive because its objective function involves large matrix multiplications (an $N \times N$ kernel matrix multiplying an $N \times N$ similarity matrix, where $N$ is the number of cells). In addition, although the learned similarity matrix could help clustering analyses, it may distort the manifold structure as demonstrated in the t-SNE plots on the learned similarity matrix[54] because the SIMLR objective function encourages forming clusters. The DeepCyTOF[55] framework has a component that uses a denoising autoencoder (trained on the cells with few or without zeros events) to filter CyTOF data to minimize the influence of dropout noises in single-cell data. The purpose of DeepCyTOF is quite different from that of scvis to model and visualize the low-dimensional structures in high-dimensional single-cell data. The most similar approach for scvis may be the parametric t-SNE algorithm[51], which uses a neural network to learn a parametric mapping from the high-dimensional space to a low dimension. However, parametric t-SNE is not a probabilistic model, the learned low-dimensional

embedding is difficult to interpret, and there are no likelihoods to quantify the uncertainty of each mapping.

In conclusion, the scvis algorithm provides a computational framework to compute low-dimensional embeddings of scRNA-seq data while preserving global structure of the high-dimensional measurements. We expect scvis to model and visualize structures in scRNA-seq data while providing new means to biologically interpretable results. As technical advances to profile the transcriptomes of large numbers of single cells further mature, we envisage that scvis will be of great value for routine analysis of large-scale, high-resolution mapping of cell populations.

## Methods

**A latent variable model of single-cell data**. We assume that the gene expression vector $\mathbf{x}_n$ of cell $n$ is a random vector and is governed by a low-dimensional latent vector $\mathbf{z}_n$. The graphical model representation of this latent variable model (with $N$ cells) is shown in Fig. 6a. The $\mathbf{x}_n$ distribution could be a complex high-dimensional distribution. We assume that it follows a Student's $t$-distribution given $\mathbf{z}_n$:

$$p(\mathbf{x}_n|\mathbf{z}_n, \boldsymbol{\theta}) = \mathcal{T}(\mathbf{x}_n|\boldsymbol{\mu}_{\boldsymbol{\theta}}(\mathbf{z}_n), \boldsymbol{\sigma}_{\boldsymbol{\theta}}(\mathbf{z}_n), \boldsymbol{\nu}) \tag{1}$$

where both $\boldsymbol{\mu}_{\boldsymbol{\theta}}(\cdot)$ and $\boldsymbol{\sigma}_{\boldsymbol{\theta}}(\cdot)$ are functions of $\mathbf{z}$ given by a neural network with parameter $\boldsymbol{\theta}$ and $\boldsymbol{\nu}$ is the degree of freedom parameter and learned from data. The marginal distribution $p(\mathbf{x}_n|\boldsymbol{\theta}) = \int p(\mathbf{x}_n|\mathbf{z}_n, \boldsymbol{\theta})p(\mathbf{z}_n|\boldsymbol{\theta})d\mathbf{z}_n$ can model a complex high-dimensional distribution.

We are interested in the posterior distribution of the low-dimensional latent variable given data: $p(\mathbf{z}_n | \mathbf{x}_n, \boldsymbol{\theta})$, which is intractable to compute. To approximate the posterior, we use the variational distribution $q(\mathbf{z}_n | \mathbf{x}_n, \boldsymbol{\phi}) = \mathcal{N}(\boldsymbol{\mu}_{\boldsymbol{\phi}}(\mathbf{x}_n), \text{diag}(\boldsymbol{\sigma}_{\boldsymbol{\phi}}(\mathbf{x}_n)))$ (Fig. 6b). Both $\boldsymbol{\mu}_{\boldsymbol{\phi}}(\cdot)$ and $\boldsymbol{\sigma}_{\boldsymbol{\phi}}(\cdot)$ are functions of $\mathbf{x}$ through a neural network with parameter $\boldsymbol{\phi}$. Although the number of latent variables grows with the number of cells, these latent variables are governed by a neural network with a fixed set of parameters $\boldsymbol{\phi}$. Therefore, even for datasets with large number of cells, we still can efficiently infer the posterior distributions of latent variables. The model coupled with the variational inference is called the variational autoencoder[56, 57].

Now the problem is to find the variational parameter $\boldsymbol{\phi}$ such that the approximation $q(\mathbf{z}_n | \mathbf{x}_n, \boldsymbol{\phi})$ is as close as possible to the true posterior distribution $p(\mathbf{z}_n | \mathbf{x}_n, \boldsymbol{\theta})$. The quality of the approximation is measured by the Kullback–Leibler (

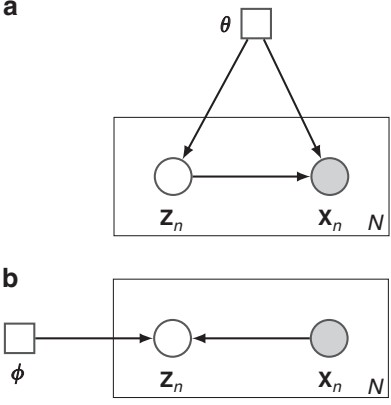

**Fig. 6** The scvis directed probabilistic graphical model and the variational approximation of its posterior. Circles represent random variables. Squares represent deterministic parameters. Shaded nodes are observed, and unshaded nodes are hidden. Here we use the plate notation, i.e., nodes inside each box will get repeated when the node is unpacked (the number of repeats is on the bottom right corner of each box). Each node and its parents constitute a family. Given the parents, a random variable is independent of the ancestors. Therefore, the joint distribution of all the random variables is the product of the family conditional distributions. **a** The generative model to generate data $\mathbf{x}_n$, and **b** the variational approximation $q(\mathbf{z}_n | \mathbf{x}_n, \boldsymbol{\phi})$ to the posterior $p(\mathbf{z}_n | \mathbf{x}_n, \boldsymbol{\theta})$

$\mathbb{KL}$) divergence[58]

$$\mathbb{KL}(q(\mathbf{z}_n|\mathbf{x}_n,\boldsymbol{\phi})||p(\mathbf{z}_n|\mathbf{x}_n,\boldsymbol{\theta}))$$
$$= \int q(\mathbf{z}_n|\mathbf{x}_n,\boldsymbol{\phi})\log\frac{q(\mathbf{z}_n|\mathbf{x}_n,\boldsymbol{\phi})}{p(\mathbf{z}_n|\mathbf{x}_n,\boldsymbol{\theta})}\,\mathrm{d}\mathbf{z}_n$$
$$= \int q(\mathbf{z}_n|\mathbf{x}_n,\boldsymbol{\phi})\log\frac{q(\mathbf{z}_n|\mathbf{x}_n,\boldsymbol{\phi})p(\mathbf{x}_n|\boldsymbol{\theta})}{p(\mathbf{z}_n,\mathbf{x}_n|\boldsymbol{\theta})}\,\mathrm{d}\mathbf{z}_n \qquad (2)$$
$$= \mathbb{E}_{q(\mathbf{z}_n|\mathbf{x}_n,\boldsymbol{\phi})}[\log q(\mathbf{z}_n|\mathbf{x}_n,\boldsymbol{\phi})]$$
$$- \mathbb{E}_{q(\mathbf{z}_n|\mathbf{x}_n,\boldsymbol{\phi})}[\log p(\mathbf{z}_n,\mathbf{x}_n|\boldsymbol{\theta})] + \log p(\mathbf{x}_n|\boldsymbol{\theta})$$

$$= \mathbb{KL}[q(\mathbf{z}_n|\mathbf{x}_n,\boldsymbol{\phi})||p(\mathbf{z}_n|\boldsymbol{\theta})]$$
$$- \mathbb{E}_{q(\mathbf{z}_n|\mathbf{x}_n,\boldsymbol{\phi})}[\log p(\mathbf{x}_n|\mathbf{z}_n,\boldsymbol{\theta})] + \log p(\mathbf{x}_n|\boldsymbol{\theta}) \qquad (3)$$

The term $\mathbb{E}_{q(\mathbf{z}_n|\mathbf{x}_n,\varphi)}[\log p(\mathbf{z}_n,\mathbf{x}_n|\boldsymbol{\theta})] - \mathbb{E}_{q(\mathbf{z}|\mathbf{x}_n,\boldsymbol{\theta})}[\log q(\mathbf{z}_n|\mathbf{x}_n,\boldsymbol{\phi})]$ in Eq. (2) is the evidence lower bound (ELBO) because it is a lower bound of $\log p(\mathbf{x}_n|\boldsymbol{\theta})$ as the $\mathbb{KL}$ divergence on the left hand side is non-negative. We therefore can do maximum-likelihood estimation of both $\boldsymbol{\theta}$ and $\boldsymbol{\phi}$ by maximizing the ELBO. Notice that in the Bayesian setting, the ELBO is a lower bound of the evidence $\log p(\mathbf{x}_n)$ as the parameters $\boldsymbol{\theta}$ are also latent random variables.

Both the prior $p(\mathbf{z}_n|\boldsymbol{\theta})$ and the variational distribution $q(\mathbf{z}_n|\mathbf{x}_n,\boldsymbol{\phi})$ in the ELBO of the form in Eq. (3) are distributions of $\mathbf{z}_n$. In our case, we can compute the $\mathbb{KL}$ term analytically because the prior is a multivariate normal distribution, and the variational distribution is also a multivariate normal distribution given $\mathbf{x}_n$. However, typically there is no closed-form expression for the integration $\mathbb{E}_{q(\mathbf{z}_n|\mathbf{x}_n,\boldsymbol{\phi})}[\log p(\mathbf{x}_n|\mathbf{z}_n,\boldsymbol{\theta})]$ because we should integrate out $\mathbf{z}_n$ and the parameters of the model $\boldsymbol{\mu}_{\boldsymbol{\theta}}(\mathbf{z}_n)$ and $\mathrm{diag}(\boldsymbol{\sigma}_{\boldsymbol{\theta}}(\mathbf{z}_n))$ are functions of $\mathbf{z}_n$. Instead, we can use Monte Carlo integration and obtain the estimated evidence lower bound for the $n$th cell:

$$\mathrm{ELBO}_n = -\mathbb{KL}(q(\mathbf{z}_n|\mathbf{x}_n,\boldsymbol{\phi})||p(\mathbf{z}_n|\boldsymbol{\theta})) + \frac{1}{L}\sum_{l=1}^{L}\log p(\mathbf{x}_n|\mathbf{z}_{n,l},\boldsymbol{\theta}) \qquad (4)$$

where $\mathbf{z}_{n,l}$ is sampled from $q(\mathbf{z}_n|\mathbf{x}_n,\boldsymbol{\phi})$ and $L$ is the number of samples. We want to take the partial derivatives of the ELBO w.r.t. the variational parameter $\boldsymbol{\phi}$ and the generative model parameter $\boldsymbol{\theta}$ to find a local maximum of the ELBO. However, if we directly sample points from $q(\mathbf{z}_n|\mathbf{x}_n,\boldsymbol{\phi})$, it is impossible to use the chain rule to take the partial derivative of the second term of Eq. (4) w.r.t $\boldsymbol{\phi}$ because $\mathbf{z}_{n,l}$ is a number. To use gradient-based methods for optimization, we indirectly sample data from $q(\mathbf{z}_n|\mathbf{x}_n,\boldsymbol{\phi})$ using the "reparameterization trick"[56, 57]. Specifically, we first sample $\boldsymbol{\varepsilon}_l$ from a easy to sample distribution $\boldsymbol{\epsilon}_l \sim p(\boldsymbol{\epsilon}|\boldsymbol{\alpha})$, e.g., a standard multivariate Gaussian distribution for our case. Next we pass $\boldsymbol{\epsilon}_l$ through a continuous function $g_{\boldsymbol{\phi}}(\boldsymbol{\epsilon},\mathbf{x}_n)$ to get a sample from $q(\mathbf{z}_n|\mathbf{x}_n,\boldsymbol{\phi})$. For our case, if $q(\mathbf{z}_n|\mathbf{x}_n,\boldsymbol{\phi}) = \mathcal{N}(\boldsymbol{\mu}_{\varphi}(\mathbf{x}_n),\,\mathrm{diag}\,(\boldsymbol{\sigma}_{\boldsymbol{\phi}}(\mathbf{x}_n)))$, then $g_{\boldsymbol{\phi}}(\boldsymbol{\epsilon},\mathbf{x}_n) = \boldsymbol{\mu}_{\boldsymbol{\phi}}(\mathbf{x}_n) + \mathrm{diag}(\boldsymbol{\sigma}_{\boldsymbol{\phi}}(\mathbf{x}_n)) \times \boldsymbol{\epsilon}$.

**Adding regularizers on the latent variables**. Given *i.i.d* data $\mathcal{D} = \{\mathbf{x}_n\}_{n=1}^{N}$, by maximizing the $\sum_n \mathrm{ELBO}_{n=1}^{N}$, we can do maximum-likelihood estimation of the model parameters $\boldsymbol{\theta}$ and the variational distribution parameters $\boldsymbol{\phi}$. Although $p(\mathbf{z}_n|\boldsymbol{\theta})p(\mathbf{x}_n|\mathbf{z}_n,\boldsymbol{\theta})$ may model the data distribution very well, the variational distribution $q(\mathbf{z}_n|\mathbf{x}_n,\boldsymbol{\phi})$ is not necessarily good for visualization purposes. Specifically, it is possible that there are no very clear gaps among the points from different clusters. In fact, to model the data distribution well, the low-dimensional $\mathbf{z}$ space tends to be filled such that all the $\mathbf{z}$ space is used in modeling the data distribution. To better visualize the manifold structure of a dataset, we need to add regularizers to the objective function in Eq. (4) to encourage forming gaps between clusters and at the same time keeping nearby points in the high-dimensional space nearby in the low-dimensional space. Here we use the non-symmetrized t-SNE[34–39] objective function.

The t-SNE algorithm preserves the local structure in the high-dimensional space after dimension reduction. To measure the "localness" of a pairwise distance, for a data point $i$ in the high-dimensional space, the pairwise distance between $i$ and another data point $j$ is transformed to a conditional distribution by centering an isotropic univariate Gaussian distribution at $i$

$$p_{j|i} = \frac{\exp\left(-\|\mathbf{x}_i - \mathbf{x}_j\|^2/2\sigma_i^2\right)}{\sum_{k\neq i}\exp\left(-\|\mathbf{x}_i - \mathbf{x}_k\|^2/2\sigma_i^2\right)} \qquad (5)$$

The point-specific standard deviation $\sigma_i$ is a parameter that is computed automatically in such a way that the perplexity ($2^{-\sum_j p_{j|i}\log_2 p_{j|i}}$) of the conditional distribution $p_{j|i}$ equals a user defined hyperparameter (e.g., typically 30[48]). We set $p_{i|i} = 0$ because only pairwise similarities are of interest.

In the low-dimensional space, the conditional distribution $q_{j|i}$ is defined similarly and $q_{i|i}$ is set to 0. The only difference is that an unscaled univariate Student's $t$-distribution is used instead of an isotropic univariate Gaussian distribution as in the high-dimensional space. Because in the high-dimensional space more points can be close to each other than in the low-dimensional space (e.g., only two points can be mutually equidistant in a line, three points in a two-dimensional plane, and four points in a three-dimensional space), it is impossible to faithfully preserve the high-dimensional pairwise distance information in the low-dimensional space if the intrinsic dimensionality of the data is bigger than that

of the low-dimensional space. A heavy tailed Student's $t$-distribution allows moderate distances in the high-dimensional space to be modeled by much larger distances in the low-dimensional space to prevent crushing different clusters together in the low-dimensional space[34].

The low-dimensional embedding coordinates $\{\mathbf{z}_i\}_{i=1}^{N}$ are obtained by minimizing the $\mathbb{KL}$ divergence between the sum of conditional distributions:

$$\sum_i \mathbb{KL}\left(p_{\cdot|i}||q_{\cdot|i}\right) = \sum_{i=1}^{N}\sum_{j=1,j\neq i}^{N} p_{j|i}\log\frac{p_{j|i}}{q_{j|i}}$$
$$= \sum_{i=1}^{N}\sum_{j=1,j\neq i}^{N} p_{j|i}\log p_{j|i} - \sum_{i=1}^{N}\sum_{j=1,j\neq i}^{N} p_{j|i}\log q_{j|i}$$
$$\propto -\sum_{i=1}^{N}\sum_{j=1,j\neq i}^{N} p_{j|i}\log\frac{\left(1+\|\mathbf{z}_i-\mathbf{z}_j\|^2/\nu\right)^{-\frac{\nu+1}{2}}}{\sum_{k,k\neq i}\left(1+\|\mathbf{z}_i-\mathbf{z}_k\|^2/\nu\right)^{-\frac{\nu+1}{2}}}$$
$$\propto -\sum_{i=1}^{N}\sum_{j=1,j\neq i}^{N} p_{j|i}\log\left(1+\left\|\mathbf{z}_i-\mathbf{z}_j\right\|^2/\nu\right)^{-\frac{\nu+1}{2}}$$
$$+ \sum_{i=1}^{N}\sum_{j=1,j\neq i}^{N} p_{j|i}\log\sum_{k,k\neq i}\left(1+\|\mathbf{z}_i-\mathbf{z}_k\|^2/\nu\right)^{-\frac{\nu+1}{2}}$$
$$\propto -\sum_{i=1}^{N}\sum_{j=1,j\neq i}^{N} p_{j|i}\log\left(1+\left\|\mathbf{z}_i-\mathbf{z}_j\right\|^2/\nu\right)^{-\frac{\nu+1}{2}} \qquad (6)$$

$$+ \sum_{i=1}^{N}\log\sum_{k,k\neq i}\left(1+\mathbf{z}_i-\mathbf{z}_k^2/\nu\right)^{-\frac{\nu+1}{2}} \qquad (7)$$

Here $\nu$ is the degree of freedom of the Student's $t$-distribution, which is typically set to one (the standard Cauchy distribution) or learned from data. Equation (6) is a data-dependent term (depending on the high-dimensional data) that keeps nearby data points in the high-dimensional data nearby in the low-dimensional space[37]. Equation (7) is a data-independent term that pushes data points in the low-dimensional space apart from each other[34]. Notice that the t-SNE objection function[34] minimizes the $\mathbb{KL}$ divergence of the joint distribution defined as the symmetrized condition distributions $p_{i,j} = (p_{i|j} + p_{j|i})/(2\times N)$ and $q_{i,j} = (q_{i|j} + q_{j|i})/(2\times N)$. t-SNE has shown excellent results on many visualization tasks such as visualizing scRNA-seq data and CyTOF data[40].

The final objective function is a weighted combination of the ELBO of the latent variable model and the above asymmetric t-SNE objective function:

$$\arg\min_{\boldsymbol{\theta},\boldsymbol{\phi}}\left(-\sum_{n=1}^{N}\mathrm{ELBO}_n + \alpha\sum_{n=1}^{N}\mathbb{KL}\left(p_{\cdot|n}||q_{\cdot|n}\right)\right) \qquad (8)$$

The parameter $\alpha$ is set to the dimensionality of the input high-dimensional data because the magnitude of the log-likelihood term in the ELBO scales with the dimensionality of the input data. The perplexity parameter is set to ten for scvis.

**Sensitivity of scvis on cell numbers**. To test the performance of scvis on scRNA-seq datasets with small numbers of cells, we ran scvis on subsampled data from the bipolar dataset. The bipolar dataset consists of six batches of datasets. We only used the cells from batch six (6221 cells in total after removing cell doublets and contaminants) to remove batch effects. Specifically, we subsampled 1, 2, 3, 5, 10, 20, 30, and 50% of the bipolar dataset from batch six (62, 124, 187, 311, 622, 1244, 1866, and 3110 cells, respectively). Then we computed the principal components from the subsampled data, and ran scvis using the top 100 PCs (for the cases with $M < 100$ cells, we used the top $M$ PCs). For each subsampled dataset, we ran scvis ten times with different seeds. We used exactly the same parameter setting for all the datasets. Therefore, except for the models trained on the cases with <100 cells, all other models have the same number of parameters.

When the number of training data is small (e.g., 62 cells, 124 cells, or 187 cells), only the large clusters such as cluster one and cluster two are distinct from the rest (Supplementary Figs. 11 and 12). As we increased the number of subsampled data points, some small clusters of cells can be recovered. At 622 cells, many cell clusters can be recovered. The $K$nn classification accuracies in Supplementary Fig. 13 (trained on the two-dimensional representations of the subsampled data and tested on the two-dimensional representations of the remaining cells from batch six) shows relatively high mean accuracies of 84.4, 84.5, 84.1, and 81.2% for $K$ equals to 5, 9, 17, and 33. When we subsampled 622 cells as in Supplementary Fig. 11e, the cells in cluster 22 were not present in the 622 cells. However, when we used the model trained on these 622 cells to embed the remaining 5599 cells (6221–622), cluster 22 cells were mapped to the "correct" region that was adjacent to cluster 20 cells and bridged cluster 20 and other clusters as in Supplementary Fig. 11d, f–h. Interestingly, with smaller numbers of cells (cell numbers ≤622), $K$nn classifiers trained on the two-dimensional scvis coordinates were better than those trained using the 100 principal components (one-sample $t$-test FDR < 0.05; Supplementary Fig. 12, the red color triangles represent the $K$nn accuracy by using the original 100 PCs).

It seems that there is no noticeable overfitting even with small numbers of cells as can be seen from the two-dimensional plots from Supplementary Figs. 11 and 12 and the $K$nn classification accuracies in Supplementary Fig. 13. To decrease the possibility of overfitting, we used Student's $t$-distributions instead of Gaussian distributions for the model neural networks. In addition, we used relatively small neural networks (a three-layer inference network with 128, 64, and 32 units and a five-layer model network with 32, 32, 32, 64, and 128 units), which may decrease the chances of overfitting.

**Sensitivity of scvis on hyperparameters**. scvis has several hyperparameters such as the number of layers in the variational inference and the model neural networks, the layer sizes (the number of units in each layer), and the $\alpha$ parameters in Eq. (8). We established that scvis is typically robust to these hyperparameters. We first tested the influence of the number of layers using batch six of the bipolar dataset. We learned scvis models with different layers from the 3110 subsampled data points from the batch six bipolar dataset. Specifically, we tested these variational influence neural networks and the model neural network layer combinations (the first number is the number of layers in the variational influence neural network, and the second number is the number of layers in the model neural network): (10, 1), (10, 3), (10, 5), (10, 7), (10, 10), (7, 10), (5, 10), (3, 10), and (1, 10). These models performed reasonably well such that the cells from the same cluster are close to each other in the two-dimensional spaces (Supplementary Fig. 14). When the number of layers in the variational inference neural network was fixed to ten, for some types of cells, their two-dimensional embeddings were close to each other and formed curve-like structures as can be seen from Supplementary Fig. 14e. The reason for this phenomenon could be that the variational influence neural network underestimated the variances of the latent $\mathbf{z}$ posterior distributions or the optimization was not converged. On the contrary, when the influence networks have smaller number of layers (<10), we did not see these curve structures (Supplementary Fig. 14f–i). The out-of-sample mapping results in Supplementary Fig. 15 show similar results.

We computed the $K$nn classification accuracies of the out-of-samples. As before, the $K$nn classifiers were trained on the two-dimensional coordinates of the training subsampled data, and the classifiers were used to classify the two-dimensional coordinates of the out-of-sample data. The parameter setting did influence scvis performance, i.e., for each $K \in \{5,9,17,33,65,129,257\}$, the trained scvis models did significantly different (Supplementary Fig. 16, FDR < 0.05, one-way analysis of variance (ANOVA) test). To find out which parameter combinations led to inferior or superior performance, we then compared the classification accuracies of each model with the most complex model with both ten layers of variational influence neural networks and ten layers of model networks. The FDR (two-sided Welch's $t$-test) at the top of each subfigure of Supplementary Fig. 16 shows that, except for $K = 257$, all the models with one layer of variational influence neural networks did significantly worse than those from the most complex model (FDR < 0.05, two-sided Welch's $t$-test). Similarly, the models with three layers of variational influence neural networks did significantly worse than those from the most complex model when $K \in \{5, 9, 17, 33, 65\}$. While for other models, their performances were not statistically different from those of the most complex models.

We next examined the influence of the layer sizes of the neural networks. The number of layers was fixed at ten for both the variational influence neural networks and the model neural networks; the number of units in each layer was set to 8, 16, 32, 64, and 128. All layers of the inference and the model neural networks had the same size. All models successfully embedded both the training data and the out-of-sample test data (Supplementary Fig. 17). However, the layer size parameter did influence scvis performance, i.e., the $K$nn classifiers on the out-of-sample data did significantly different (Supplementary Fig. 18, FDR < 0.05, one-way ANOVA test). The FDR (two-sided Mann–Whitney $U$-test) at the top of each subfigure of Supplementary Fig. 18 shows that all models with layer size of eight did significantly worse than those from the most complex model using 128 units (FDR < 0.05). Similarly, the models with layer size of 16 did significantly worse than those from the most complex model when $K \in \{5, 9, 17, 33, 65, 129\}$ (FDR < 0.05). While for other models, their performances were not statistically different from those from the most complex models. Notice that, at layer size of 64, the mapping functions from one run were worse than others in embedding the out-of-sample data. However, there was no significant difference in the log-likelihoods from the repeated ten runs (Supplementary Fig. 19, one-way ANOVA $p$-value = 0.741).

For the $\alpha$ weight parameter in Eq. (8), we set $\alpha$ relative to the dimensionality of the input data. We set $\alpha = 0, 0.5, 1.0, 1.5, 2.0, 10.0,$ inf times of the dimensionality of the input data. When $\alpha = $ inf, the trained models did significantly worse than the models trained with the default $\alpha$ equals to the dimensionality of the input data for $K \in \{5, 9, 17, 33, 65\}$ (FDR ≤ 0.05, two-sided Welch's $t$-test, Supplementary Figs. 20, 21 and 22). Also, when $\alpha = 0$, the trained models were significantly worse than the models trained with the default $\alpha$ equaling to the dimensionality of the input data for all $K$s (FDR ≤ 0.05, two-sided Welch's $t$-test, Supplementary Fig. 22). For $\alpha = 0$, we performed an extra comparison by using the synthetic nine-dimensional data, showing that when $K$ was large (≥65), setting $\alpha = 9$ (the dimensionality of the input data) did significantly better than letting $\alpha = 0$ (Supplementary Fig. 23, FDR ≤ 0.05, one-sided Welch's $t$-test).

**Computational complexity analysis**. The scvis objective function involves the asymmetrical t-SNE objective function. The most time-consuming part is to compute the pairwise distances between two cells in a mini-batch that takes $O(TN^2D + TN^2d)$ time, where $N$ is the mini-batch size (we use $N = 512$ in this study), $D$ is the dimensionality of the input data, $d$ is the dimensionality of the low-dimensional latent variables (e.g., $d = 2$ for most cases), and $T$ is the number of iterations. For our case, we first use PCA to project the scRNA-seq data to a 100-dimensional space, so $D = 100$. For a feedforward neural network with $L$ hidden layers, and the number of neurons in layer $l$ is $n_l$, the time complexity to train the neural network is $O(NT \sum_{i=0}^{L} n_{l+1} * n_l)$, where $n_0 = D$ and $n_{L+1}$ is the size of the output layer. For the model neural network, we use a five hidden layer (with layer size 32, 32, 32, 64, and 128) feedforward neural network. The input layer size is $d$ and the output layer size is $D$. For the variational inference network, we use a three hidden layer (with layer size 128, 64, and 32) feedforward neural network. The size of the input layer is $D$ and the size of the output layer is $d$. For space complexity, we need to save the weights and bias of each neuron $O(\sum_{i=0}^{L} n_{l+1} * n_l)$. We also need to save the $O(N^2)$ pairwise distances and the data of size $O(ND)$ in a mini-batch.

The original t-SNE algorithm is not scalable to large datasets (with tens of thousands of cells to millions of cells) because it needs to compute the pairwise distances between any two cells (taking $O(M^2D + M^2T)$ time and $O(M^2)$ space, where $M$ is the total number of cells and $T$ is the number of iterations). Approximate t-SNE algorithms are typically more scalable in both time and space. For example, BH t-SNE only computes the distance between a cell and its $K$nns. Therefore, BH t-SNE takes $O(M \log(M))$ time and $O(M \log(M))$ space, where we assume $K$ is in the order of $O(\log(M))$.

We next experimentally compare the scalability of scvis and BH t-SNE (the widely used Rtsne package[48]) by using the 1.3 million cells from 10X genomics[59]. However, BH t-SNE did not finish in 24 h and we terminated it. On the contrary, scvis produced visually reasonable results in <33 min (after 3000 mini-batch training, Supplementary Fig. 24a). Therefore, scvis can be much more scalable than BH t-SNE for very large datasets. As we increased the number of training batches, we can see slightly better separations in clusters as in Supplementary Fig. 24b–f. The time used to train scvis increased linearly in the number of training mini-batches (Supplementary Fig. 24h). However, for small datasets, BH t-SNE can be more efficient than scvis. For example, for the melanoma dataset with only 4645 cells, scvis still took 24 min to run 3000 mini-batches, while BH t-SNE finished in only 28.9 s. All the experiments were conducted using a Mac computer with 32 GB of RAM, 4.2 GHz four-core Intel i7 processor with 8 MB cache.

Finally, when the mapping function is trained, mapping new cells takes only $O(M \sum_{l=0}^{L} n_{l+1} * n_l)$ time, where $M$ is the number of input cells. Also, because each data point can be mapped independently, the space complexity could be only $O(\sum_{l=0}^{L+1} n_{l+1} * n_l)$. As an example, it took only 1.5 s for a trained scvis model to map the entire 1.3 million cells from 10X genomics.

**Datasets**. The oligodendroglioma dataset measures the expression of 23,686 genes in 4347 cells from six *IDH1* or *IDH2* mutant human oligodendrolioma patients[44]. The expression of each gene is quantified as $\log_2 (TPM/10+1)$, where "TPM" standards for "transcripts per million"[60]. Through copy number estimations from these scRNA-seq measurements, 303 cells without detectable copy number alterations were classified as normal cells. These normal cells were further grouped into microglia and oligodendrocyte based on a set of marker genes they expressed. Two patients show subclonal copy number alterations.

The melanoma dataset is from sequencing 4645 cells isolated from 19 metastatic melanoma patients[3]. The cDNAs from each cell were sequenced by an Illumina NextSeq 500 instrument to 30 bp pair-end reads with a median of ~150,000 reads per cell. The expression of each gene (23,686 genes in total) is quantified by $\log_2 (TPM/10+1)$. In addition to malignant cells, the authors also profiled immune cells, stromal cells, and endothelial cells to study the whole-tumor multi-cellular ecosystem.

The bipolar dataset consists of low-coverage (median depth of 8200 mapped reads per cell) Drop-seq sequencing[9] of 27,499 mouse retinal bipolar neural cells from a transgenic mouse[1]. In total, 26 putative cells types were identified by clustering the first 37 principal components of all the 27,499 cells. Fourteen clusters can be assigned to bipolar cells, and another major cluster is composed of Mueller glia cells. These 15 clusters account for about 96% of all the 27,499 cells. The remaining 11 clusters (comprising of only 1060 cells) include rod photoreceptors, cone photoreceptors, amacrine cells, and cell doublets and contaminants[1].

The retina dataset consists of low-coverage Drop-seq sequencing[9] of 44,808 cells from the retinas of 14-day-old mice. By clustering the two-dimensional t-SNE embedding using DBSCAN[61]—a density-based clustering algorithm, the authors identified 39 clusters after merging the clusters without enough differentially expressed genes between any two clusters.

The 10X Genomics neural cell dataset consist of 1,306,127 cells from cortex, hippocampus, and subventricular zones of two E18 C57BL/6 mice. The cells were sequenced on 11 Illumina Hiseq 4000 machines to produce 98 bp reads[59].

For the mass cytometry dataset H1[17], manual gating assigned 72,463 cells to 14 cell types based on 32 measured surface protein markers. Manual gating assigned 31,721 cells to the same 14 cell populations from H2 based on the same 32 surface protein markers.

**Statistical analysis**. All statistical analyses were performed using the R statistical software package, version 3.4.3. Boxplots denote the medians and the interquartile ranges (IQRs). The whiskers of a boxplot are the lowest datum still within 1.5 IQR of the lower quartile and the highest datum still within 1.5 IQR of the upper quartile. The full datasets were superimposed to boxplots. For datasets with non-normal distribution (e.g., outliers), non-parametric tests were used. To account for unequal variances, Welch's *t*-test was used for pairwise data comparison. Adjusted *p*-values <0.05 (FDR, the Benjamini–Hochberg procedure[62]) were considered to be significant.

**Code availability**. The scvis v0.1.0 Python package is available freely from bitbucket: https://bitbucket.org/jerry00/scvis-dev.

**Data availability**. The scRNA-seq data that support the findings of this study are available in Gene Expression Omnibus with the identifiers (bipolar: GSE81905, retina: GSE63473, oligodendroglioma: GSE70630, metastatic melanoma: GSE72056, E18 mouse neural cells: GSE93421). The E18 mouse neural cells are freely available from 10X Genomics[59]. The other scRNA-seq data are publicly available from the single-cell portal[45]. The mass cytometric data can be downloaded from cytoback[63]. The synthetic data used in this study are available from bitbucket repo: https://bitbucket.org/jerry00/scvis-dev.

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

## Acknowledgements

This work was supported by a Discovery Frontiers project grant, "The Cancer Genome Collaboratory", jointly sponsored by the Natural Sciences and Engineering Research Council (NSERC), Genome Canada (GC), the Canadian Institutes of Health Research (CIHR), and the Canada Foundation for Innovation (CFI) to S.P.S. In addition, we acknowledge generous long-term funding support from the BC Cancer Foundation. The S.P.S. group receives operating funds from the Canadian Breast Cancer Foundation, the Canadian Cancer Society Research Institute (impact grant 701584 to S.P.S.), the Terry Fox Research Institute (grant 1021, The Terry Fox New Frontiers Program Project Grant in the Genomics of Forme Fruste Tumours: New Vistas on Cancer Biology and Treatment, and grant 1061, The Terry Fox New Frontiers Program Project Grant in Overcoming Treatment Failure in Lymphoid Cancers), CIHR (grant MOP-115170 to S.P.S.), and CIHR Foundation (grant FDN-143246 to S.P.S.). S.P. S. is supported by Canada Research Chairs. S.P.S. is a Michael Smith Foundation for Health Research scholar. S.P.S is a Susan G. Komen Scholar.

## Author contributions

J.D.: project conception, software implementation, and data analysis; J.D., A.C., and S.P.S.: algorithm development and manuscript writing; S.P.S.: project conception, oversight, and senior responsible author.

## Additional information

**Competing interests:** S.P.S is a shareholder of Contextual Genomics Inc. The remaining authors declare no competing interests.

