## [Peer Review File · Nature Communications]

Reviewers' comments:

Reviewer #1 (Remarks to the Author):

The authors developed scviz, a dimensionality reduction (DR) algorithm for single-cell RNA sequencing (scRNAseq) data and potentially other single-cell data sources (such as mass cytometry). scviz is based on a variational autoencoder (VAE), with an additional regularization term whose purpose is to encourage the formation of gaps between clumps of data points. The regularization is based on the mismatch metric from SNE, an earlier DR method. scviz appears to offer two main advantages over existing methods. One, it has an emphasis on both local and global structure. Therefore, the arrangement of clumps in the low-dimensional map is indicative of their positioning in the high-dimensional space. Two, scviz is a probabilistic generative model, allowing the calculation of log-likelihoods and the embedding of out-of-sample points in the low-dimensional map. The authors have applied the method to three snRNAseq data sets.

Validity

1) The authors have indicated several flaws with t-SNE, which is a popular method for DR. However, they did not address similar concerns with scviz:

- What are the time and space complexities of scviz, and how do they scale with method parameters? Additionally, it would be beneficial to see run time comparisons to t-SNE.
- How sensitive is scviz to the various parameters included in its operation, such as the number of layers in the neural network, the layer sizes, the alpha parameter used in weighing the SNE mismatch metric or the perplexity parameter used in calculating it? (The authors state that "the perplexity parameter for scvis was stable for different numbers of input data points" but provide no data, and do not discuss other parameters that are involved in their model.)

2) The authors benchmark scviz against t-SNE using a simulated data set. However, they did not benchmark the embedding of out-of-sample data. Two possible algorithms to test against are PCA and parameteric t-SNE. While the authors mention possible pitfalls of parameteric t-SNE in their discussion, these are irrelevant for the purpose of the kNN preservation score that they use, so the method could be compared against.

3) The authors do not benchmark the embedding of out-of-sample data in scRNAseq data, only in their simulated data set. While they do embed the whole mouse retina data set using the mapping function from the bipolar data, which is qualitatively sound, they are missing a quantitative analysis (perhaps by separating a single data set into training and testing).

4) The main contribution of the authors over the base VAE is the regularization term. However, there is no evidence showing that the regularization term is even necessary. The authors should provide "before-and-after" figures of the same data set (preferably both synthetic and scRNAseq data), with and without the regularization.

Originality and significance

To the best of my knowledge this is the first use of VAEs in the DR of single-cell data. DeepCyTOF is an automatic gating method for cytometry data that utilizes an autoencoder (the manuscript is only available on bioRxiv). However, it learns cell labeling, not a low-dimensional embedding. Parametric t-SNE (van der Maaten, 2009) utilizes a feed-forward neural network, but it learns an embedding that was generated by another method (namely, t-SNE). Therefore, it is my opinion

that the scviz method is original.

With that said, the contribution detailed in the manuscript is relatively limited: as far as I can see, the VAE the authors use is an "out-of-the-box" formulation and implementation, with very little in the way of innovation. The one exception is the SNE regularization term, which is adopted from a previous method and whose contribution is unclear (as explained above).

From a significance perspective, I appreciate scviz's strengths of detecting global structure and providing a mapping function. However, my personal opinion is that neither offer significant breakthroughs in this field. One, the global structure of single-cell data sets is highly complex and it is debatable whether the two-dimensional representation chosen by a given scviz run is biologically relevant. Two, the utility of the mapping function is diminished given the fact that both PCA and the BH implementation of t-SNE can process large data sets relatively quickly and efficiently, and that the technical variation between samples that were acquired over long time periods would be detrimental to the accuracy of the dimensionality reduction anyway.

From a biological perspective, the method has been applied to previously published data sets, with no new data. The manuscript includes little to no new biological insight. There is a passing mention of a "mixture cell population" which is "substantially different from other bipolar cells" in the retina dataset, but no further characterization of that population. Additionally, under the melanoma data set, there is a brief discussion of whether fibroblasts are more similar to healthy cells or malignant cells, but no additional exploration.

Data & methodology

The quality of data and presentation is valid for the analyses and benchmarks that are discussed in the manuscript. The methods section includes a thorough discussion of the method details. The authors have supplied the code necessary to run the method and the data for the bipolar and whole retina data sets.

I would prefer if the authors to provided the synthetic data set discussed in figures 1 and 2, or at least the code necessary to generate a similar data set.

Appropriate use of statistics and treatment of uncertainties

Legends, legend labels, and axis labels are missing throughout the manuscript. In general, while the figures include all necessary information, they do not follow standard formatting practices. The authors should revise all of the figures to make sure that all required labels and explanations are included, either in the figure itself, the figure legend, or both.

Using just one figure as an example, in figure 1 a-c, the figure should indicate that the colors represent different clusters. In figure 1 d, the figure is missing a legend of log-likelihood values. In figure 1 f-g, the figure is missing an explanation of the statistics used (I am guessing that these are standard boxplot statistics, but that should be indicated in the figure legend, which currently only says "mean"). Also, I am surprised that in figure g, all values correspond to integer values -- my understanding is that these are mean values over many points, and as such I expect them to be float values.

These issues persist in all of the figures, including in the supplement.

Conclusions

The authors' conclusions and data interpretation are valid. With that said, the method and results are limited, as discussed above.

Suggested improvements

Setting aside the issues mentioned above, there are three areas where I would want to see the applicability of scviz:

- 1) The method does not address one of the greatest issues with t-SNE and related methods: multiple runs over the same data set lead to different maps, and there is no way to compare maps between runs or data sets. I suspect that this is one area where scviz's mapping function could be of benefit.
- 2) The authors only utilize scviz in the context of two-dimensional maps. How does the method perform when reducing the data to three-dimensional maps?
- 3) The authors use a synthetic data set and three scRNAseq data sets. I am curious to see how the method performs over flow and mass cytometry data.

Minor notes

- In page 3, the authors state that "we add the t-SNE objective function on the latent z distribution as a constraint". However, looking at the methods (page 16), equation 6 is sum over i for $KL(p_{\cdot|i} || q_{\cdot|i})$. This is the objective function for the original SNE, not t-SNE (see equation 2 in van der Maaten and Hinton, 2008). The objective function for t-SNE is the symmetric version of the above. If the authors use the former, they should update the text to "the SNE objective function". If the latter, they should update the equation.

Reviewer #2 (Remarks to the Author):

This manuscript presents a deep-learning based dimension reduction method for single cell data and demonstrates its advantage over tSNE.

Comments:

1. The method was mainly evaluated in contrast to tSNE, it would help to bring in other methods such as GPLVM
2. Deep-learning relies on big data. Does the method need a lot of cells to train the network? When the cell number is low, do we encounter over-fitting issue? It would help to suggest the minimum number of cells that suits this method and can train a neural network that can be representative and generalized enough to predict new data.
3. Does the method favor bigger clusters more compared to smaller clusters? In the presence of rare cells, can the method uncover them from other more abundant cells?
4. Training a large neural network with multiple layers and a number of nodes per layer is time-

consuming. How long does this method take for the analysis that was presented in the manuscript?

5. How was the number of layers and number of nodes per layer determined? Is it data specific? Does it affect the results if these numbers were changed?

6. How is the method different from the original tensorflow for embedding? Any specific adjustment was made so that it is applicable for single cell dimension reduction?

7. TensorFlow/deep learning/neural network is not straightforward to most readers. It would help to add a schematic figure in main text to illustrate how it works and add a few lines to describe it in a more lay-man language.

8. How is KNN trained to predict embedding? Can this be used to predict new data?

9. Page 6 paragraph 2 line 7, Figure 2(a-h) should be Supplementary Figure 2(a-h)

10. In Figure 2d, what is the perplexity used for t-SNE analysis? Will the performance improve if we increase the perplexity?

11. When a model or neural network is trained on dataset A, and then is applied to dataset B, can it discover new clusters that are only present in dataset B but not A. Can the trained model predict new clusters in unseen data?

12. How do we compare the results from 1) training directly on dataset B 2) and train on dataset A but predict on dataset B. Are these two results similar or very different? And if we 3) train the model on combined dataset A and dataset B (i.e. A+B), would the embedding be more comprehensive to uncover all different clusters. It would be helpful to discuss the pro and cons of these possible options.

Please find enclosed our point-by-point responses to the reviewer comments in *blue italic font*.

Reviewer #1 (Remarks to the Author): The authors developed scviz, a dimensionality reduction (DR) algorithm for single-cell RNA sequencing (scRNAseq) data and potentially other single-cell data sources (such as mass cytometry). scviz is based on a variational autoencoder (VAE), with an additional regularization term whose purpose is to encourage the formation of gaps between clumps of data points. The regularization is based on the mismatch metric from SNE, an earlier DR method. scviz appears to offer two main advantages over existing methods. One, it has an emphasis on both local and global structure. Therefore, the arrangement of clumps in the low-dimensional map is indicative of their positioning in the high-dimensional space. Two, scviz is a probabilistic generative model, allowing the calculation of log-likelihoods and the embedding of out-of-sample points in the low-dimensional map. The authors have applied the method to three scRNAseq data sets.

We really appreciate the reviewer for the positive feedback and detailed comments on how to improve the manuscript. We have responded the reviewer's comments as follows.

Validity

1) The authors have indicated several flaws with t-SNE, which is a popular method for DR. However, they did not address similar concerns with scviz:

- What are the time and space complexities of scviz, and how do they scale with method parameters? Additionally, it would be beneficial to see run time comparisons to t-SNE.

We thank the reviewer for the suggestion. We have analyzed the time and space complexity of scvis and their comparison with those of t-SNE and BH t-SNE. We also experimentally show that scvis is more scalable than BH t-SNE for vary large datasets such as the 1.3 million neural cells for E18 mice from 10X genomics. The results are presented in the Methods section (Computational complexity analysis, pp. 22-23) as follows:

*"The scvis objective function involves the asymmetrical t-SNE objective function. The most time-consuming part is to compute the pairwise distances between two cells in a mini-batch which takes $O(TN^2D + TN^2d)$ time, where N is the mini-batch size (we use $N=512$ in this study), and D is the dimensionality of the input data, d is the dimensionality of the low-dimensional latent variables (e.g., $d=2$ for most cases), and T is the number of iterations. For our case, we first use PCA to project the scRNA-seq data to a 100-dimensional space, so $D=100$. For a feedforward neural network with L hidden layers, and the number of neurons in layer l is n_l , the time complexity to train the neural network is $O(NT \sum_{i=0}^L n_{i+1} * n_i)$, where $n_0 = D$ and n_{L+1} is the size of the output layer. For the model neural network, we use a five hidden layer (with layer size 32,32,32,64,and 128) feedforward neural network. The input layer size is d and the output layer size is D . For the variational inference network, we use a three-hidden layer (with layer size 128,64,and 32) feedforward neural network. The size of the input layer is D and the size of the*

Re: revision of manuscript NCOMMS-17-17768A: “Interpretable dimensionality reduction of single cell transcriptome data with deep generative models”.

output layer is d . For space complexity, we need to save the weights and bias of each neuron $O(\sum_{i=0}^L n_{i+1} * n_i)$. We also need to save the $O(N^2)$ pairwise distances and the data of size $O(ND)$ in a mini-batch.

The original t-SNE algorithm is not scalable to large datasets (with tens of thousands of cells to millions of cells) because it needs to compute the pairwise distances between any two cells (taking $O(M^2D + M^2T)$ time and $O(M^2)$ space, where M is the total number of cells, and T is the number of iterations). Approximate t-SNE algorithms are typically more scalable in both time and space. For example, BH t-SNE only computes the distance between a cell and its K -nearest neighbors. Therefore, BH t-SNE takes $O(M\log(M))$ time and $O(M\log(M))$ space, where we assume K is in the order of $O(\log(M))$.

We next experimentally compare the scalability of scvis and BH t-SNE (the widely used Rtsne package) by using the 1.3 million cells from 10X genomics. However, BH t-SNE did not finish in 24 hours and we terminated it. On the contrary, scvis produced visually reasonable results in less than 33 minutes (after 3,000 mini-batch training, Supplementary Fig. 24(a)). Therefore, scvis can be much more scalable than BH t-SNE for very large datasets. As we increased the number of training batches, we can see slightly better separations in clusters as in Supplementary Fig. 24(b-f). The time used to train scvis increased linearly in the number of training mini-batches (Supplementary Fig. 24(h)). However, for small datasets, BH t-SNE can be more efficient than scvis. For example, for the melanoma dataset with only 4,645 cells, scvis still took 24 minutes to run 3,000 mini-batches, while BH t-SNE finished in only 28.9 seconds. All the experiments were conducted using a Mac computer with 32 GB of RAM, 4.2 GHz four-core Intel i7 processor with 8 MB cache.

Finally, when the mapping function is trained, mapping new cells takes only $O(M \sum_{i=0}^L n_{i+1} * n_i)$ time, where M is the number of input cells. Also, because each data point can be mapped independently, the space complexity could be only $O(\sum_{i=0}^{L+1} n_{i+1} * n_i)$. As an example, it took only 1.5 seconds for a trained scvis model to map the whole 1.3 million cells from 10X genomics.”

- How sensitive is scviz to the various parameters included in its operation, such as the number of layers in the neural network, the layer sizes, the alpha parameter used in weighing the SNE mismatch metric or the perplexity parameter used in calculating it? (The authors state that “the perplexity parameter for scvis was stable for different numbers of input data points” but provide no data, and do not discuss other parameters that are involved in their model.)

Following the reviewer’s suggestion, we systematically tested the sensitivity of scvis to the number of layers in the neural networks, the layer sizes, and the alpha parameters used in weighing the SNE. The results are presented in the Methods section (Sensitivity of scvis on hyper-parameters), pp. 20-22:

“scvis has several hyper-parameters such as the number of layers in the variational inference and the model neural networks, the layer sizes (the number of units in each layer), and the α parameters in Equation 8. We established that scvis is typically robust to these hyper-parameters. We first tested the influence of the number of layers using batch six of the bipolar dataset. We learned scvis models with different layers from the 3,110 subsampled data points from the batch six bipolar dataset. Specifically, we tested these variational influence neural networks and the model neural network layer combinations (the first number is the number of layers in the variational influence neural network, and the second number is the number of layers in the model neural network): (10, 1), (10, 3), (10, 5), (10, 7), (10, 10), (7, 10), (5, 10), (3, 10), and (1, 10). These models performed reasonably well such that the cells from the same cluster are close to each other in the two-dimensional space (Supplementary Fig. 14). When the number of layers in the variational inference neural network was fixed to ten, for some types of cells, their two dimensional embeddings were close to each other and formed curve like structures as can be seen from Supplementary Fig. 14 (e). The reason for this phenomenon could be that the variational influence network under-estimated the variances of the latent z posterior distributions. On the contrary, when the influence networks have smaller number of layers (<10), we didn’t see

Re: revision of manuscript NCOMMS-17-17768A: “Interpretable dimensionality reduction of single cell transcriptome data with deep generative models”.

these curve structures (Supplementary Fig. 14 (f-i)). The out-of-sample mapping results in Supplementary Fig. 15 show similar results.

We computed the Knn classification accuracies of the out-of-samples. As before, the Knn classifiers were trained on the two-dimensional coordinates of the training subsampled data, and the classifiers were used to classify the two-dimensional coordinates of the out-of-sample data. The parameter setting did influence scvis performance, i.e., for each $K \in \{5,9,17,33,65,129,257\}$, the trained scvis models did significantly different (Supplementary Fig.16, FDR <0.05, one-way ANOVA test). To find out which parameter combinations lead to inferior performance, we then compared the classification accuracies of each model with the most complex model with both ten layers of variational influence neural networks, and ten layers of model networks. The FDR (two-sided Welch’s t-test) at the top of each subfigure of Supplementary Fig.16 shows that except for $K=257$, all the models with one layer of variational influence neural networks did significantly worse than those from the most complex model (FDR < 0.05, two-sided Welch’s t-test). Similarly, the models with three layers of variational influence neural networks did significantly worse than those from the most complex model when $K \in \{5,9,17,33, 65\}$. While for other models, their performances were not statistically different from those of the most complex models.

We next examined the influence of the layer sizes of the neural networks. The number of layers was fixed at ten for both the variational influence neural networks and the model neural networks; the number of units in each layer was set to 8, 16, 32, 64, and 128. All layers of the inference and the model neural networks had the same size. All models successfully embedded both the training data and the out-of-sample test data (Supplementary Fig.17). However, the layer size parameter did influence scvis performance, i.e., the Knn classifiers on the out-of-sample data did significantly different (Supplementary Fig. 18, FDR <0.05, one-way ANOVA test). The FDR (Mann-Whitney U-test) at the top of each subfigure of Supplementary Fig. 18 shows that all the models with layer size of eight did significantly worse than those from the most complex model using 128 units. Similarly, the models with layer size of 16 did significantly worse than those from the most complex model when $K \in \{5,9,17,33, 65, 129\}$. While for other models, their performances were not statistically different from those from the most complex models. Notice that at layer size of 64, the mapping functions from one run were worse than others in embedding the out-of-sample data. However, there was no significant difference in the log-likelihoods from the repeated runs (Supplementary Fig. 19, one way ANOVA test p-value = 0.741).

For the α weight parameter in Equation 8, we set α relative to the dimensionality of the input data. We set $\alpha=0,0.5,1.0,1.5,2.0,10.0, \text{inf}$ times of the dimensionality of the input data. When $\alpha=\text{inf}$, the trained models did significantly worse than the models trained with the default α equals to the dimensionality of the input data for $K \in \{5,9,17,33,65\}$ (Supplementary Fig. 20-22). Also, when $\alpha=0$, the trained models were significantly worse than the models trained with the default α equals to the dimensionality of the input data for all Ks (FDR ≤ 0.05 , two-sided Welch’s t-test). For $\alpha=0$, we performed an extra comparison by using the synthetic nine-dimensional data (Supplementary Fig. 23), showing that when K is large (≥ 65), setting $\alpha=9$ (the dimensionality of the input data) was significantly better than $\alpha=0$ (FDR < 0.05, one-sided Welch’s t-test).”

For the perplexity parameter, typically for larger datasets, the perplexity should increase accordingly. For scvis, we do mini-batch optimization by sampling a mini-batch of cells (e.g., 512 cells) each time. Therefore, for a given mini-batch size, scvis is less sensitive to the perplexity as the total number of training points changes. We have revised our sentences accordingly in pp. 8-9 as follows:

“In contrast, as we adopted mini-batch for training scvis by subsampling, e.g., 512 cells each time, scvis was less sensitive to the perplexity parameter as we increase the total number of training data points because the number of cell is fixed at 512 at each training step. Therefore, scvis performed well on approximately an order of magnitude larger dataset (Fig. 3(d-e)), without

Re: revision of manuscript NCOMMS-17-17768A: "Interpretable dimensionality reduction of single cell transcriptome data with deep generative models".

changing the perplexity parameter for scvis. For this larger dataset, the t-SNE results (Fig. 3(f)) were difficult to interpret without the ground-truth cluster information because it was already difficult to see how many clusters in this dataset, not to mention to uncover the overall structure of the data. Although by increasing the perplexity parameter, t-SNE performed better (Supplementary Fig. 4), the outliers still formed distinct clusters, and it remains difficult to set this parameter in practice."

2) The authors benchmark scviz against t-SNE using a simulated data set. However, they did not benchmark the embedding of out-of-sample data. Two possible algorithms to test against are PCA and parameteric t-SNE. While the authors mention possible pitfalls of parameteric t-SNE in their discussion, these are irrelevant for the purpose of the kNN preservation score that they use, so the method could be compared against.

Following the reviewer's suggestion, we benchmarked scvis against PCA, parametric t-SNE (pt-SNE), and GPLVM for embedding out-of-sample data (the synthetic dataset), and comparing the Knn classification accuracies for different Ks for each method. The results are presented in the Results section (p. 7) as follows:

"We then benchmarked scvis against Gaussian process latent variate model (GPLVM, implemented in the GPy package), parametric t-SNE (pt-SNE), and PCA on embedding the 22,000 out-of-sample data points. We used the eleven scvis models trained on the small nine-dimensional synthetic dataset with 2,200 data points to embed the larger nine-dimensional synthetic data with 22,000 data points. Similarly, we trained eleven GPLVM models and pt-SNE models on the small nine-dimensional synthetic dataset, and applied these models to the bigger synthetic dataset. To compare the abilities of the trained models to embed unseen data, we trained K-nearest neighbor classifiers on the two-dimensional representations (of the small 2,200 data points) outputted from different algorithms. These Knn classifiers were used to classify the two-dimensional coordinates of the 22,000 data points outputted from different algorithms. scvis was significantly better than GPLVM and pt-SNE for different Ks (Fig. 3(c), FDR < 0.05, one-sided Mann-Whitney U-test). For PCA, because the model is unique for a given dataset, we generated unique two-dimensional coordinates for the 22,000 out-of-sample data points. The Knn classifiers trained on the PCA coordinates were worse than those from scvis, GPLVM, and pt-SNE in terms of the mean classification accuracies for different Ks."

3) The authors do not benchmark the embedding of out-of-sample data in scRNAseq data, only in their simulated data set. While they do embed the whole mouse retina data set using the mapping function from the bipolar data, which is qualitatively sound, they are missing a quantitative analysis (perhaps by separating a single data set into training and testing).

Following the reviewer's suggestion, we then benchmarked scvis against PCA, parametric t-SNE, and GPLVM on the bipolar dataset. The results are presented in the Results section (pp. 9-11) as follows:

"We benchmarked scvis against GPLVM, pt-SNE, PCA on embedding out-of-sample scRNA-seq data. We did a five-fold cross-validation analysis on the bipolar dataset. Specifically, we portioned the bipolar dataset into five roughly equal size subsamples, and held out one subsample as out-of-sample evaluation data, using the remaining four subsamples as training data to learn different models. We then trained Knn classifiers on the two-dimensional representations of the training data, and then used the Knn classifiers to classify the two-dimensional representations of the out-of-sample evaluation data. The process was repeated five times with each of the five subsamples used exactly once as the out-of-sample validation data. scvis was significantly better than pt-SNE, GPLVM, and PCA on embedding the out-of-samples (Supplementary Fig. 6(a-b), FDR < 0.05, one-sided Welch's t-test)."

Re: revision of manuscript NCOMMS-17-17768A: “Interpretable dimensionality reduction of single cell transcriptome data with deep generative models”.

4) The main contribution of the authors over the base VAE is the regularization term. However, there is no evidence showing that the regularization term is even necessary. The authors should provide “before-and-after” figures of the same data set (preferably both synthetic and scRNAseq data), with and without the regularization.

Following the reviewer’s suggestion, we ran scvis by removing the t-SNE regularizer on the bipolar scRNA-seq dataset and the synthetic dataset. The results are presented in the Methods section: Sensitivity of scvis on hyper-parameters, p. 22, paragraph 1:

“Also, when $\alpha = 0$, the trained models were significantly worse than the models trained with the default α equaling to the dimensionality of the input data for all Ks (FDR ≤ 0.05 , two-sided Welch’s t-test, Supplementary Fig. 22). For $\alpha = 0$, we did an extra comparison by using the synthetic nine-dimensional data, showing that when K is large (≥ 65), setting $\alpha = 9$ (the dimensionality of the input data) did significantly better than letting $\alpha = 0$ (Supplementary Fig. 23, FDR <0.05 , one-sided Welch’s t-test).”

Originality and significance

To the best of my knowledge this is the first use of VAEs in the DR of single-cell data. DeepCyTOF is an automatic gating method for cytometry data that utilizes an autoencoder (the manuscript is only available on bioRxiv). However, it learns cell labeling, not a low-dimensional embedding. Parametric t-SNE (van der Maaten, 2009) utilizes a feed-forward neural network, but it learns an embedding that was generated by another method (namely, t-SNE). Therefore, it is my opinion that the scviz method is original.

We thank the reviewer for considering scvis as original. We also thank the reviewer for pointing us to these papers. We briefly discuss both paper in the Discussion section (p. 15):

“The DeepCyTOF framework has a component that uses a denoising autoencoder (trained on the cells with few or without zeros events) to filter CyTOF data to minimize the influence of dropout noises in single-cell data. The purpose of DeepCyTOF is quite different from that of scvis to model and visualize the low-dimensional structure in high-dimensional single-cell data. The most similar approach to scvis may be the parametric t-SNE algorithm, which uses a neural network to learn a parametric mapping from the high-dimensional space to a low dimension. However, parametric t-SNE is not a probabilistic model, the learned low-dimensional embedding is difficult to interpret, and there is no likelihoods to quantify the uncertainty of each mapping.”

With that said, the contribution detailed in the manuscript is relatively limited: as far as I can see, the VAE the authors use is an “out-of-the-box” formulation and implementation, with very little in the way of innovation. The one exception is the SNE regularization term, which is adopted from a previous method and whose contribution is unclear (as explained above).

In this manuscript, we developed scvis to model and visualize the low-dimensional structures in single-cell datasets. Although both VAE and t-SNE are powerful tools and have been used successfully in many fields, they are typically used independently. Here we first showed that we can simultaneously learn a good low-dimensional representation of the high-dimensional single-cell data (from the inference network) and a good model of the high-dimensional data such that the data have high log-likelihoods. We showed that scvis had several advantages over t-SNE. First, it naturally learned an inference network to map out-of-samples; Second, it better preserved the global structures in the high-dimensional data that could be useful for us to investigate the biology; third, the log-likelihoods added rich information for us to quantify the uncertainty of the embedding of each data point; And finally, scvis was scalable to really large datasets for which BH t-SNE is too slow to run.

Re: revision of manuscript NCOMMS-17-17768A: “Interpretable dimensionality reduction of single cell transcriptome data with deep generative models”.

From a significance perspective, I appreciate scviz’s strengths of detecting global structure and providing a mapping function. However, my personal opinion is that neither offer significant breakthroughs in this field. One, the global structure of single-cell data sets is highly complex and it is debatable whether the two-dimensional representation chosen by a given scviz run is biologically relevant. Two, the utility of the mapping function is diminished given the fact that both PCA and the BH implementation of t-SNE can process large data sets relatively quickly and efficiently, and that the technical variation between samples that were acquired over long time periods would be detrimental to the accuracy of the dimensionality reduction anyway.

We thank the reviewer for recognizing the strength of scvis in recovering the structures in scRNA-seq data. We agree with the reviewer that it’s really hard to faithfully preserve the global structures of high-dimensional datasets. However, we do think that preserving some global structures of the original data will greatly help us to better interpret the data and form better hypothesis. For example, if we see two cell types are very close to each other, we may infer that these cells could be similar in cell lineage, or communicate with each other.

Although other dimension-reduction methods such as BH-tsne can process large dataset, the inherent limitations of BH-tsne still makes it hard to interpret the results, for example, both compact clusters and outliers in the high-dimensional space can form compact clusters in the low-dimensional t-SNE plots. Moreover, for large datasets, e.g., the 10X Genomics E18 mice dataset with about 1.3 million cells, BH t-SNE is not scalable and scvis can produce visually reasonable results in less than 33 minutes (pp. 22-23, Computational complexity analysis).

We can use the model trained on one large dataset, and only need to fine-tune the model on additional datasets, which is fast and keeps the advantages of scvis.

From a biological perspective, the method has been applied to previously published data sets, with no new data. The manuscript includes little to no new biological insight. There is a passing mention of a “mixture cell population” which is “substantially different from other bipolar cells” in the retina dataset, but no further characterization of that population. Additionally, under the melanoma data set, there is a brief discussion of whether fibroblasts are more similar to healthy cells or malignant cells, but no additional exploration.

Although we only analyzed publically available data in this manuscript, as scRNA-seq technologies mature, and ambitious international projections such as the Human Cell Atlas (Regev et al, 2017) to produce large-scale single cell data, we think that the computational method development for scRNA-seq analysis has fallen behind data generation capacity. It is of course imperative to maximize ability to extract biological information from these datasets. Our focus in this paper is provide the community with an approach to increase capacity for biological knowledge generation with an enhanced method. We leave it to future work to apply scvis to analyze large-scale scRNA-seq data for new biological insights, as this will need extensive wet-lab experiments to confirm and prove the findings: work that is clearly beyond the scope of this submission.

Data & methodology

The quality of data and presentation is valid for the analyses and benchmarks that are discussed in the manuscript. The methods section includes a thorough discussion of the method details. The authors have supplied the code necessary to run the method and the data for the bipolar and whole retina data sets.

We thank the reviewer for the positive comments. We have made the code public to allow users to analyze their scRNA-seq data (<https://bitbucket.org/jerry00/scvis-dev>).

Re: revision of manuscript NCOMMS-17-17768A: "Interpretable dimensionality reduction of single cell transcriptome data with deep generative models".

I would prefer if the authors to provided the synthetic data set discussed in figures 1 and 2, or at least the code necessary to generate a similar data set.

Following the reviewer's suggestion, we added the synthetic data to our repo:
<https://bitbucket.org/jerry00/scvis-dev>

Appropriate use of statistics and treatment of uncertainties

Legends, legend labels, and axis labels are missing throughout the manuscript. In general, while the figures include all necessary information, they do not follow standard formatting practices. The authors should revise all of the figures to make sure that all required labels and explanations are included, either in the figure itself, the figure legend, or both.

Using just one figure as an example, in figure 1 a-c, the figure should indicate that the colors represent different clusters. In figure 1 d, the figure is missing a legend of log-likelihood values. In figure 1 f-g, the figure is missing an explanation of the statistics used (I am guessing that these are standard boxplot statistics, but that should be indicated in the figure legend, which currently only says "mean"). Also, I am surprised that in figure g, all values correspond to integer values -- my understanding is that these are mean values over many points, and as such I expect them to be float values.

These issues persist in all of the figures, including in the supplement.

We really appreciate the reviewer for pointing out these issues. We updated all the figures and figure legends (both main figures and supplementary figures), and accordingly, the figure presentation has been improved. For Fig. 1(g), before only one run results were presented. This has been augmented with a presentation of ten repeated runs, thereby generating better results.

Conclusions

The authors' conclusions and data interpretation are valid. With that said, the method and results are limited, as discussed above.

We appreciate the reviewer for considering our conclusions and data interpretation as valid. Our method first showed that we can simultaneously learn a good low-dimensional representation of the high-dimensional single-cell data and a good model of the high-dimensional data that with good log-likelihoods. To model noisy single-cell data, we used robust Student's t model as opposed to conventional multivariate Gaussian model to model the data distribution.

Our results on both synthetic data and real single-cell data showed that 1) scvis outperformed competitive methods in embedding out-of-sample data; 2) it preserved the global structures in the high-dimensional data better than t-SNE; 3) the log-likelihood added rich information for us to quantify the uncertainty of the embedding of each data point; 4) we finally showed that scvis is scalable to currently large datasets that BH t-SNE is too slow to run. Also, it is very efficient to map out-of-sample data.

Suggested improvements

Setting aside the issues mentioned above, there are three areas where I would want to see the applicability of scviz:

1) The method does not address one of the greatest issues with t-SNE and related methods: multiple runs over the same data set lead to different maps, and there is no way to compare maps between runs or data sets. I suspect that this is one area where scviz's mapping function could be of benefit.

Re: revision of manuscript NCOMMS-17-17768A: “Interpretable dimensionality reduction of single cell transcriptome data with deep generative models”.

We thank the reviewer for the suggestion. Although multiple runs of scvis produce visually similar results (Supplementary Fig. 1(a-j)), some mapping functions do better for embedding new data. For example, as in Supplementary Fig.18, with layer size 64, the mapping function from one run (with random seed 3) was worse than those from other runs. Therefore, with a validation dataset, we can possibly detect those less accurate mapping functions. However, from the estimated log-likelihoods we cannot determine which mapping function is worse (Supplementary Fig. 19). We thus leave using the likelihood function for model selection for future investigations.

2) The authors only utilize scviz in the context of two-dimensional maps. How does the method perform when reducing the data to three-dimensional maps?

We appreciate the reviewer’s suggestion. In our original submission, we only reported the two-dimensional embedding for better visualization in print because some points could be shaded when we examine the map from only one viewpoint for three-dimensional scatter plots. For the software, the users can set the dimensionality of the low-dimensional representation.

To evaluate how scvis performs on three-dimensional maps, we project the bipolar data to a three-dimensional space. The results are presented in the Results section (p. 12):

“It is straightforward to project scRNA-seq to a higher than two-dimensional space. To evaluate how scvis performs on higher-dimensional maps, we projected the bipolar data to a three-dimensional space. We obtained better average log-likelihood per data point, i.e., 255.1 versus 253.3 (from the last 100 iterations) by projecting the data to a three-dimensional space compared to projecting the data to a two-dimensional space (Supplementary Fig. 7). In addition, the average KL divergence was smaller (2.7 versus 4.1 from the last 100 iterations) by projecting the data to a three-dimensional space.”

3) The authors use a synthetic data set and three scRNAseq data sets. I am curious to see how the method performs over flow and mass cytometry data.

To test the performance of scvis on other single-cell datasets, we used the mass-cytometry data from Levine et al (2015). This dataset is from a healthy adult donor H2. Manual gating assigned 31,721 cells to 14 cell types based on 32 surface protein markers. The results are presented in the Results section (p. 12):

“Finally, to demonstrate that scvis can be used for other types of single-cell data, we learned a parametric mapping from the CyTOF data H2, and then directly used the mapping to project the CyTOF data H1 to a two-dimensional space. All the 14 cell types were separated (although CD16+ and CD16- NK cells have some overlaps), and CD4 T-cells and CD8 T-cells clusters are adjacent to each other. Moreover, the high quality of the mapping carried over to the CyTOF data H1 (72,463 cells) (Supplementary Fig.8 (a,b)).”

Minor notes

- In page 3, the authors state that “we add the t-SNE objective function on the latent z distribution as a constraint”. However, looking at the methods (page 16), equation 6 is sum over i for $KL(p_{\{i\}} || q_{\{i\}})$. This is the objective function for the original SNE, not t-SNE (see equation 2 in van der Maaten and Hinton, 2008). The objective function for t-SNE is the symmetric version of the above. If the authors use the former, they should update the text to “the SNE objective function”. If the latter, they should update the equation.

We thank the reviewer for pointing this out. In our implementation, we used the asymmetric version of the t-SNE objective function. We therefore updated our text as follows (p. 19):

Re: revision of manuscript NCOMMS-17-17768A: “Interpretable dimensionality reduction of single cell transcriptome data with deep generative models”.

*“Notice that the t-SNE objection function minimizes the KL divergence of the joint distribution defined as the symmetrized condition distributions $p_{i,j} = (p_{ij} + p_{ji})/(2 * N)$ and $q_{i,j} = (q_{ij} + q_{ji})/(2 * N)$ ”*

“The final objective function is a weighted combination of the ELBO of the latent variable model and the above asymmetric t-SNE objective function”

Reviewer #2 (Remarks to the Author): This manuscript presents a deep-learning based dimension reduction method for single cell data and demonstrates its advantage over tSNE.

Comments:

1. The method was mainly evaluated in contrast to tSNE, it would help to bring in other methods such as GPLVM

Following the reviewer’s suggestion, we have benchmarked scvis against GPLVM, parametric-tSNE, and PCA for dimension reduction. The results are presented in the Results section (pp. 9-11):

“We benchmarked scvis against GPLVM, pt-SNE, PCA on embedding out-of-sample scRNA-seq data, performing a five-fold cross-validation analysis on the bipolar dataset. Specifically, we partitioned the bipolar dataset into five roughly equal size subsamples, and held out one subsample as out-of-sample evaluation data, using the remaining four subsamples as training data to learn different models. We then trained Knn classifiers on the two-dimensional representations of the training data, and used the Knn classifiers to classify the two-dimensional representations of the out-of-sample evaluation data. The process was repeated five times with each of the five subsamples used exactly once as the out-of-sample validation data. scvis was significantly better than pt-SNE, GPLVM, and PCA on embedding the out-of-samples (Supplementary Fig. 6(a-b), FDR < 0.05, one-sided Welch’s t-test).”

2. Deep-learning relies on big data. Does the method need a lot of cells to train the network? When the cell number is low, do we encounter over-fitting issue? It would help to suggest the minimum number of cells that suits this method and can train a neural network that can be representative and generalized enough to predict new data.

Following the reviewer’s suggestion, we examine the sensitivity of scvis on the cell numbers by down-sampling analysis. The results are presented in the Methods section (Sensitivity of scvis on cell numbers, pp. 19-20):

“To test the performance of scvis on scRNA-seq datasets with small numbers of cells, we ran scvis on subsampled data from the bipolar dataset. The bipolar dataset consists of six batches of datasets. We only used the cells from batch six (6,221 cells in total after removing cell doublets and contaminants) to remove batch effects. Specifically, we subsampled 1%, 2%, 3%, 5%, 10%, 20%, 30%, and 50% percent of the bipolar dataset from batch six (62, 124, 187, 311, 622, 1,244, 1,866, and 3,110 cells, respectively). Then we computed the principal components from the subsampled data, and run scvis using the top 100 PCs (for the cases with $M < 100$ cells, we used the top M PCs). For each subsampled dataset, we ran scvis ten times with different seeds. We used exactly the same parameter setting for all the datasets. Therefore, except for the models trained on the cases with less than 100 cells, all other models have the same number of parameters.

When the number of training data is small (e.g., 62 cells, 124 cells, or 187 cells), only the large clusters such as cluster one and cluster two are distinct from the rest (Supplementary Fig. 11-12). As we increased the number of subsampled data points, some small clusters of cells can be

Re: revision of manuscript NCOMMS-17-17768A: “Interpretable dimensionality reduction of single cell transcriptome data with deep generative models”.

recovered. At 622 cells, many cell clusters can be recovered. The Knn classification accuracies in Supplementary Fig. 13 (trained on the two-dimensional representations of the subsampled data and tested on the two-dimensional representations of the remaining cells from batch six) shows relatively high mean accuracies of 84.4%, 84.5%, 84.1%, and 81.2% for K equals to 5, 9, 17, and 33. When we subsampled 622 cells as in Supplementary Fig.11 (e), the cells in cluster 22 were not present in the 622 cells. However, when we used the model trained on these 622 cells to embed the remaining 5,599 cells (6,221 - 622), cluster 22 cells were mapped to the ‘correct’ region that was adjacent to cluster 20 cells and bridged cluster 20 and other clusters as in Supplementary Fig. 11 (d,f-h). Interestingly, with smaller numbers of cells (cell numbers ≤ 622), K-nearest neighbor classifiers trained on the two-dimensional scvis coordinates were better than those trained using the 100 principal components (one sample t-test FDR < 0.05 ; Supplementary Fig. 12, the red color triangles represent the Knn accuracy by using the original 100 PCs).

There was no noticeable overfitting even with small numbers of cells as can be seen from the two-dimensional plots from Supplementary Fig. 11, Supplementary Fig. 12, and the Knn classification accuracies in Supplementary Fig. 13. To decrease the possibility of overfitting, we used Student’s t distributions instead of Gaussian distributions for the model neural networks. In addition, we used relatively small neural networks (a three-layer inference network with 128, 64, 32 units and a five-layer model network with 32, 32, 32, 64, 128 units) which may decrease the chances of overfitting.”

3. Does the method favor bigger clusters more compared to smaller clusters? In the presence of rare cells, can the method uncover them from other more abundant cells?

The method puts most of its power to model the big clusters to get high overall log-likelihoods. As can be seen from Fig. 1(d) and Fig. 3(b), the cells in ‘big’ clusters, or in the set of major clusters (bipolar cells in Fig.3) tend to have high likelihoods. The minor clusters (non-bipolar cells in Fig. 3) still form distinct clusters but have low likelihoods.

4. Training a large neural network with multiple layers and a number of nodes per layer is time-consuming. How long does this method take for the analysis that was presented in the manuscript?

We thank the reviewer for pointing out this important issue. We have analyzed the time and space complexity of scvis and their comparison with those of t-SNE and BH t-SNE. We also experimentally show that scvis is more scalable than BH t-SNE for vary large datasets such as the 1.3 million neural cells for E18 mice from 10X genomics. The results are presented in the Methods section (Computational complexity analysis, pp. 22-23)

“The scvis objective function involves the asymmetrical t-SNE objective function. The most time-consuming part is to compute the pairwise distances between two cells in a mini-batch which takes $O(TN^2D + TN^2d)$ time, where N is the mini-batch size (we use $N=512$ in this study), and D is the dimensionality of the input data, d is the dimensionality of the low-dimensional latent variables (e.g., $d=2$ for most cases), and T is the number of iterations. For our case, we first use PCA to project the scRNA-seq data to a 100-dimensional space, so $D=100$. For a feedforward neural network with L hidden layers, and the number of neurons in layer l is n_l , the time complexity to train the neural network is $O(NT \sum_{i=0}^L n_{i+1} * n_i)$, where $n_0 = D$ and n_{L+1} is the size of the output layer. For the model neural network, we use a five hidden layer (with layer size 32,32,32,64,and 128) feedforward neural network. The input layer size is d and the output layer size is D . For the variational inference network, we use a three-hidden layer (with layer size 128,64,and 32) feedforward neural network. The size of the input layer is D and the size of the output layer is d . For space complexity, we need to save the weights and bias of each neuron $O(\sum_{i=0}^L n_{i+1} * n_i)$. We also need to save the $O(N^2)$ pairwise distances and the data of size $O(ND)$ in a mini-batch.

Re: revision of manuscript NCOMMS-17-17768A: “Interpretable dimensionality reduction of single cell transcriptome data with deep generative models”.

The original t-SNE algorithm is not scalable to large datasets (with tens of thousands of cells to millions of cells) because it needs to compute the pairwise distances between any two cells (taking $O(M^2D + M^2T)$ time and $O(M^2)$ space, where M is the total number of cells, and T is the number of iterations). Approximate t-SNE algorithms are typically more scalable in both time and space. For example, BH t-SNE only computes the distance between a cell and its K -nearest neighbors. Therefore, BH t-SNE takes $O(M\log(M))$ time and $O(M\log(M))$ space, where we assume K is in the order of $O(\log(M))$.

We next experimentally compare the scalability of scvis and BH t-SNE (the widely used Rtsne package) by using the 1.3 million cells from 10X genomics. However, BH t-SNE did not finish in 24 hours and we terminated it. On the contrary, scvis produced visually reasonable results in less than 33 minutes (after 3,000 mini-batch training, Supplementary Fig. 24(a)). Therefore, scvis can be much more scalable than BH t-SNE for very large datasets. As we increased the number of training batches, we can see slightly better separations in clusters as in Supplementary Fig. 24(b-f). The time used to train scvis increased linearly in the number of training mini-batches (Supplementary Fig. 24(h)). However, for small datasets, BH t-SNE can be more efficient than scvis. For example, for the melanoma dataset with only 4,645 cells, scvis still took 24 minutes to run 3,000 mini-batches, while BH t-SNE finished in only 28.9 seconds. All the experiments were conducted using a Mac computer with 32 GB of RAM, 4.2 GHz four-core Intel i7 processor with 8 MB cache.

Finally, when the mapping function is trained, mapping new cells takes only $O(M \sum_{i=0}^L n_{i+1} * n_i)$ time, where M is the number of input cells. Also, because each data point can be mapped independently, the space complexity could be only $O(\sum_{i=0}^{L+1} n_{i+1} * n_i)$. As an example, it took only 1.5 seconds for a trained scvis model to map the whole 1.3 million cells from 10X genomics.”

5. How was the number of layers and number of nodes per layer determined? Is it data specific? Does it affect the results if these numbers were changed?

We did not do extensive parameter optimization in our original submission. Typically, the method is robust to different hyper-parameters. We have now investigated the impacts of parameters on scvis performance, and the results are presented in the Methods section (Sensitivity of scvis on hyper-parameters, pp. 20 - 22):

“scvis has several hyper-parameters such as the number of layers in the variational inference and the model neural networks, the layer sizes (the number of units in each layer), and the α parameters in Equation 8. We established that scvis is typically robust to these hyper-parameters. We first tested the influence of the number of layers using batch six of the bipolar dataset. We learned scvis models with different layers from the 3,110 subsampled data points from the batch six bipolar dataset. Specifically, we tested these variational influence neural networks and the model neural network layer combinations (the first number is the number of layers in the variational influence neural network, and the second number is the number of layers in the model neural network): (10, 1), (10, 3), (10, 5), (10, 7), (10, 10), (7, 10), (5, 10), (3, 10), and (1, 10). These models performed reasonably well such that the cells from the same cluster are close to each other in the two-dimensional space (Supplementary Fig. 14). When the number of layers in the variational inference neural network was fixed to ten, for some types of cells, their two dimensional embeddings were close to each other and formed curve like structures as can be seen from Supplementary Fig. 14 (e). The reason for this phenomenon could be that the variational influence network under-estimated the variances of the latent z posterior distributions. On the contrary, when the influence networks have smaller number of layers (<10), we didn't see these curve structures (Supplementary Fig. 14 (f-i)). The out-of-sample mapping results in Supplementary Fig. 15 show similar results.

We computed the Knn classification accuracies of the out-of-samples. As before, the Knn classifiers were trained on the two-dimensional coordinates of the training subsampled data, and the classifiers were used to classify the two-dimensional coordinates of the out-of-sample data.

The parameter setting did influence scvis performance, i.e., for each $K \in \{5,9,17,33,65,129,257\}$, the trained scvis models did significantly different (Supplementary Fig.16, $FDR < 0.05$, one-way ANOVA test). To find out which parameter combinations lead to inferior performance, we then compared the classification accuracies of each model with the most complex model with both ten layers of variational influence neural networks, and ten layers of model networks. The FDR (two-sided Welch’s t-test) at the top of each subfigure of Supplementary Fig.16 shows that except for $K=257$, all the models with one layer of variational influence neural networks did significantly worse than those from the most complex model ($FDR < 0.05$, two-sided Welch’s t-test). Similarly, the models with three layers of variational influence neural networks did significantly worse than those from the most complex model when $K \in \{5,9,17,33, 65\}$. While for other models, their performances were not statistically different from those of the most complex models.

We next examined the influence of the layer sizes of the neural networks. The number of layers was fixed at ten for both the variational influence neural networks and the model neural networks; the number of units in each layer was set to 8, 16, 32, 64, and 128. All layers of the inference and the model neural networks had the same size. All models successfully embedded both the training data and the out-of-sample test data (Supplementary Fig.17). However, the layer size parameter did influence scvis performance, i.e., the Knn classifiers on the out-of-sample data did significantly different (Supplementary Fig.18, $FDR < 0.05$, one-way ANOVA test). The FDR (Mann-Whitney U-test) at the top of each subfigure of Supplementary Fig. 18 shows that all the models with layer size of eight did significantly worse than those from the most complex model using 128 units. Similarly, the models with layer size of 16 did significantly worse than those from the most complex model when $K \in \{5,9,17,33, 65, 129\}$. While for other models, their performances were not statistically different from those from the most complex models. Notice that at layer size of 64, the mapping functions from one run were worse than others in embedding the out-of-sample data. However, there was no significant difference in the log-likelihoods from the repeated runs (Supplementary Fig. 19, one way ANOVA test p -value = 0.741).

For the α weight parameter in Equation 8, we set α relative to the dimensionality of the input data. We set $\alpha=0,0.5,1.0,1.5,2.0,10.0$, inf times of the dimensionality of the input data. When $\alpha=inf$, the trained models did significantly worse than the models trained with the default α equals to the dimensionality of the input data for $K \in \{5,9,17,33,65\}$ (Supplementary Fig. 20-22). Also, when $\alpha=0$, the trained models were significantly worse than the models trained with the default α equals to the dimensionality of the input data for all K s ($FDR \leq 0.05$, two-sided Welch’s t-test). For $\alpha=0$, we performed an extra comparison by using the synthetic nine-dimensional data (Supplementary Fig. 23), showing that when K is large (≥ 65), setting $\alpha=9$ (the dimensionality of the input data) was significantly better than $\alpha=0$ ($FDR < 0.05$, one-sided Welch’s t-test).”

For the perplexity parameter, typically for larger datasets, the perplexity should increase accordingly. For scvis, we do mini-batch optimization by sampling a mini-batch of cells (e.g., 512 cells) each time. Therefore, for a given mini-batch size, scvis is less sensitive to the perplexity as the total number of training points changes. We have revised our sentences accordingly in pp. 8-9 as follows:

“In contrast, as we adopted mini-batch for training scvis by subsampling, e.g., 512 cells each time, scvis was less sensitive to the perplexity parameter as we increase the total number of training data points because the number of cell is fixed at 512 at each training step. Therefore, scvis performed well on approximately an order of magnitude larger dataset (Fig. 3(d-e)), without changing the perplexity parameter for scvis. For this larger dataset, the t-SNE results (Fig. 3(f)) were difficult to interpret without the ground-truth cluster information because it was already difficult to see how many clusters in this dataset, not to mention to uncover the overall structure of the data. Although by increasing the perplexity parameter, t-SNE performed better (Supplementary Fig. 4), the outliers still formed distinct clusters, and it remains difficult to set this parameter in practice.”

Re: revision of manuscript NCOMMS-17-17768A: “Interpretable dimensionality reduction of single cell transcriptome data with deep generative models”.

6. How is the method different from the original tensorflow for embedding? Any specific adjustment was made so that it is applicable for single cell dimension reduction?

If we understand the question correctly, the tensorflow embedding projector has the standard t-SNE and PCA for dimension reduction. Both tools are general tools can be used for other data types. Although we implemented scvis using tensorflow python APIs, we did not use the tensorflow embedding projector.

7. TensorFlow/deep learning/neural network is not straightforward to most readers. It would help to add a schematic figure in main text to illustrate how it works and add a few lines to describes it in a more lay-man language.

We really appreciate the reviewer for this suggestion. In the revised version of the manuscript, we add Fig.1 to illustrate how the model works (p. 4):

“Overview of the scvis method. (a) scvis model assumptions: given a low-dimensional point drawn from a simple distribution, e.g., a two-dimensional standard normal distribution, a high-dimensional gene expression vector of a cell can be generated by drawing a sample from the distribution $p(x | z, \theta)$. The heatmap represents a cell-gene expression matrix, where each row is a cell, and each column is a gene. Color encodes the expression levels of genes in cells. The data-point specific parameters θ are determined by a model neural network. The model neural network (a feedforward neural network) consists of an input layer, several hidden layers, and an output layer. The output layer outputs the parameters θ of $p(x | z, \theta)$. (b) scvis inference: given a high-dimensional gene expression vector of a cell (a row of the heatmap), scvis obtains its low-dimensional representation by sampling from the conditional distribution $q(z | x, \varphi)$. The data-point specific parameters φ are determined by a variational inference neural network. The inference neural network is also a feedforward neural network and its output layer outputs the parameters φ of $q(z | x, \varphi)$. Again, the heatmap represents a cell-gene expression matrix. The scatter plot shows samples drawn from the variational posterior distributions $q(z | x, \varphi)$.”

8. How is KNN trained to predict embedding? Can this be used to predict new data?

In the manuscript, KNN was used in two ways to measure the qualities of the projections. We apologize for the confusion.

First, we use KNN as a validation metrics, to measures the neighbor preservations in the low-dimensional space. Specifically, we compute the k-nearest neighbors of each cells in the original high dimensional space and the low-dimensional embedded space, and compute the overlap of the k-nearest neighbors of each cell (we called this metric KNN preservations).

Second, when using a pre-trained model to project out-of-sample data, KNN classifiers were trained on the low-dimensional representations of the training data, and the classifiers were used to predict the labels of the low-dimensional representations of the test data (the low-dimensional representations of these out-of-sample data points were obtained by using the trained mapping functions). If the model is good for embedding out-of-samples, the KNN classifiers should have high test accuracies.

9. Page 6 paragraph 2 line 7, Figure 2(a-h) should be Supplementary Figure 2(a-h)

We thank the reviewer for pointing out the typo. We have corrected the typo.

Re: revision of manuscript NCOMMS-17-17768A: “Interpretable dimensionality reduction of single cell transcriptome data with deep generative models”.

10. In Figure 2d, what is the perplexity used for t-SNE analysis? Will the performance improve if we increase the perplexity?

In our original study, we used the default perplexity parameter of 30. Although by increasing the perplexity parameter, t-SNE performed better (Supplementary Fig. 4), the outliers still formed distinct clusters, and it's difficult to set this parameter in practice.

11. When a model or neural network is trained on dataset A, and then is applied to dataset B, can it discover new clusters that are only present in dataset B but not A. Can the trained model predict new clusters in unseen data?

Unseen clusters were likely to be mapped to some distinct regions not occupied by the trained data, and also they tend to have lower log-likelihoods (Supplementary Fig.5 (d)). So scvis results will help us to discover new clusters. However, we should notice that data from different unseen clusters, especially some sub-cluster that are 'close' to each other (e.g., different American cells in Supplementary Fig. 5 (f)) could be mapped to the same region, so we may need to re-analyze these data with low log-likelihoods.

Another example is from down-sampling analysis of the bipolar dataset (the Sensitivity of scvis on cell numbers section, p. 20, paragraph 2):

“When we subsampled 622 cells as in Supplementary Fig.10 (e), the cells in cluster 22 were not present in the 622 cells. However, when we used the model trained on these 622 cells to embed the remaining 5,599 cells (6,221 - 622), cluster 22 cells were mapped to the 'correct' region that was adjacent to cluster 20 cells and bridged cluster 20 and other clusters as in Supplementary Fig.10 (d,f-h).”

12. How do we compare the results from 1) training directly on dataset B 2) and train on dataset A but predict on dataset B. Are these two results similar or very different? And if we 3) train the model on combined dataset A and dataset B (i.e. A+B), would the embedding be more comprehensive to uncover all different clusters. It would be helpful to discuss the pro and cons of these possible options.

This could be quite data and application dependent.

Without a training dataset A, we have no choice but training on dataset B directly. If we have a training dataset A with comprehensive cell information, e.g., for the bipolar dataset, it has well annotated bipolar cells. The full retina dataset, however, has less bipolar cells. Therefore, when training on the bipolar dataset and tested on the retina dataset, we can recover more bipolar clusters than directly training on the retina dataset (p. 11, paragraph 2):

“Although Macosko et al only identified eight subtypes of bipolar cells, all the recently identified 14 subtypes of bipolar cells were possibly present in the retina dataset as can be seen from Fig. 3(c), i.e., cluster 27 (BC3B and BC4), cluster 28 (BC2 and BC3A), cluster 29 (BC1A and BC1B), cluster 30 (BC5A and BC5D), cluster 31 (BC5B and BC5C), and cluster 33 (BC6 and BC8/9).”

Although the cells only in dataset B are likely to be mapped to a unique region by using the mapping function from dataset A, it's possible that some clusters of cells that are close to each other are mapped to the same region, e.g., the different Amacrine cells were mapped to the same region (Supplementary Fig. 5(f)). So if we want to detect Amacrine cell subtypes, it's better to train directly on the retina dataset.

Re: revision of manuscript NCOMMS-17-17768A: “Interpretable dimensionality reduction of single cell transcriptome data with deep generative models”.

If the number of cells is small (e.g., less than 622 cells as in our down-sampling analysis), it’s better to combine all the data together to produce a comprehensive map.

Training on dataset B directly will produce an embedding that is different from the embedding from training on dataset A but predicting on dataset B. This could be problematic in certain situations. For example, if we have time-course data from a patient, and the embedding at each time is quite different from each other. This could cause trouble for people, e.g., a pathologist to interpret the results. Therefore, the mapping function from dataset A could be useful for this situation.

References

Levine, J. H., Simonds, E. F., Bendall, S. C., Davis, K. L., El-ad, D. A., Tadmor, M. D., ... & Finck, R. (2015). Data-driven phenotypic dissection of AML reveals progenitor-like cells that correlate with prognosis. *Cell*, 162(1), 184-197.

Regev, A., Teichmann, S., Lander, E. S., Amit, I., Benoist, C., Birney, E., ... & Clevers, H. (2017). The Human Cell Atlas. *bioRxiv*, 121202.

REVIEWERS' COMMENTS:

Reviewer #1 (Remarks to the Author):

The reviewer would like to thank the authors for their extensive updates to the manuscript. All of my comments have been addressed.

A few minor points:

- Typo, Figure 1, "given a low-dimensional point DRAWN from a simple distribution".
- Figure 2, thank you for updating the figure legend with the panel descriptions. Please add relevant labels to the panels themselves. For example, panel b should be titled "t-SNE", panel c "scviz", "FDR" to the numbers at the top of panel g, etc.
- Figure 3, same remarks, please label the components of the figure. It is cumbersome to match the legend to the panels. These same comments are applicable to all of the figures.
- In addition, the panel labels and plot tick labels are small and difficult to read. Please increase the font size throughout the manuscript.

Overall, I agree that the authors are presenting a novel, interesting method that would serve users in the high-dimensional single cell research community.

Reviewer #2 (Remarks to the Author):

The authors have responded to all comments in great details and revised the manuscript accordingly. As the single-cell techniques advance and more data are being generated by human cell atlas, we shall need new dimension reduction tools in addition to tSNE. This paper presents a new framework of using artificial neural network for single cell dimension reduction. I have no further comments

Re: revision of manuscript NCOMMS-17-17768B: "Interpretable dimensionality reduction of single cell transcriptome data with deep generative models".

Please find enclosed our point-by-point responses to the reviewer comments in *blue italic font*.

REVIEWERS' COMMENTS:

Reviewer #1 (Remarks to the Author):

The reviewer would like to thank the authors for their extensive updates to the manuscript. All of my comments have been addressed.

We are very pleased to hear that the reviewer is satisfied with our revision.

A few minor points:

- Typo, Figure 1, "given a low-dimensional point DRAWN from a simple distribution".

We appreciate the reviewer for pointing out this typo, and we corrected the typo accordingly.

- Figure 2, thank you for updating the figure legend with the panel descriptions. Please add relevant labels to the panels themselves. For example, panel b should be titled "t-SNE", panel c "scviz", "FDR" to the numbers at the top of panel g, etc.

We thank the reviewer for the suggestion, and we added labels to panels.

- Figure 3, same remarks, please label the components of the figure. It is cumbersome to match the legend to the panels. These same comments are applicable to all of the figures.

We thank the reviewer for the suggestion, and we added labels to panels.

- In addition, the panel labels and plot tick labels are small and difficult to read. Please increase the font size throughout the manuscript.

We increased the font size of panel labels and plot tick labels of figures.

Overall, I agree that the authors are presenting a novel, interesting method that would serve users in the high-dimensional single cell research community.

We thank the reviewer for taking the time to review our submission and for the suggestions that greatly improve the quality of our manuscript.

Re: revision of manuscript NCOMMS-17-17768B: "Interpretable dimensionality reduction of single cell transcriptome data with deep generative models".

Reviewer #2 (Remarks to the Author):

The authors have responded to all comments in great details and revised the manuscript accordingly. As the single-cell techniques advance and more data are being generated by human cell atlas, we shall need new dimension reduction tools in addition to tSNE. This paper presents a new framework of using artificial neural network for single cell dimension reduction. I have no further comments

We really appreciate the recognition of our work by the reviewer.